**Global submesoscale Transition Scale estimation using alongtrack satellite altimetry**

**Oscar Vergara[1,2], Rosemary Morrow[2], Marie-Isabelle Pujol[1], Gérald Dibarboure[3] &**

**Clément Ubelmann[4]**

[1] CLS Space Oceanography, Ramonville Saint-Agne, France

[2] Université de Toulouse, LEGOS (CNES/CNRS/IRD/UPS), Toulouse, France

[3] CNES, Toulouse, France

[4] Datlas, Grenoble, France

Corresponding author: Oscar Vergara (overgara@groupcls.com)

**Key Points:**

- Spatial scale of the transition from geostrophically balanced to unbalanced motions is estimated regionally from satellite altimetry data for the first time.
- Results agree with in situ observations and predictions from high-resolution models including tidal forcing.

**Abstract**

The ocean's sea surface height (SSH) field is a complex mix of motions in geostrophic balance and unbalanced motions including high-frequency tides, internal tides and internal gravity waves. Barotropic tides are well estimated for altimetric SSH in the open ocean, but the SSH signals of internal tides remains. The transition scale, Lt, at which these unbalanced ageostrophic motions dominate balanced geostrophic motions, is estimated for the first-time using satellite altimetry. Lt is critical to define the spatial scales above which surface geostrophic currents can be inferred from SSH gradients. We use a statistical approach based on the analysis of 1 Hz altimetric SSH wavenumber spectra to obtain four geophysical parameters that vary regionally and seasonally: the background error, the spectral slope in the mesoscale range, a second spectral slope at smaller scales, and Lt. The mesoscale slope and error levels are similar to previous studies based on satellite altimetry. The break in the wavenumber spectra to a flatter spectral slope can only be estimated in mid-latitude regions where the signal exceeds the altimetric noise level. Small values of Lt are observed in regions

of energetic mesoscale activity, while larger values are observed towards low latitudes and

regions of lower mesoscale activity. These results are consistent with recent analyses of in

situ observations and high-resolution models. Limitations of our results and implications for

reprocessed nadir and future swath altimetric missions are discussed.

## 1 Introduction

Global maps of multi-mission satellite altimetry sea surface height (SSH) are widely used in

the ocean community, resolving the larger mesoscale dynamic scales greater than 150-200 km

in wavelength (Chelton et al., 2011; Ballarotta et al., 2019). Our understanding of upper ocean

dynamics in the smaller mesoscale to submesoscale wavelength range (roughly 15-200 km)

has seen great improvement in recent years due to the combined use of in situ measurements

and state-of-the-art high-resolution ocean models (Sasaki et al 2014; Rocha et al. 2016a,

2016b; Qiu et al. 2017, 2018; Klein et al., 2019).   Processes at these spatial scales are

essential in determining the upper ocean energy budget through the kinetic energy cascade

and energy dissipation (e.g. Ferrari and Wunsch 2009; McWilliams, 2016; Rocha et al.,

2016a). Additionally, they play a critical role in connecting the surface ocean with the

interior, through the modulation of the mixed layer seasonality and heat transfer (Capet et al.,

2008; Klein et al., 2008; Thomas et al., 2008; Su et al., 2020; Siegelman, 2020).

Kinetic energy and SSH variance at these 15-200 km spatial scales is partitioned between

balanced (geostrophic) and unbalanced (ageostrophic) motions. Quantifying the relative

importance of each component of the flow across the ocean is capital for the correct

calculation of geostrophic currents from SSH for all satellite altimetry missions, including the

upcoming Surface Water and Ocean Topography (SWOT) high-resolution altimetry mission.

Barotropic tides are well estimated for altimetric SSH in the open ocean, but the SSH signals

of other ageostrophic high-frequency motions remains. Recent results show that, depending

on the location and season, the energy and SSH signature associated with unbalanced motions (including near-inertial flows, internal tides, and inertia-gravity waves) can overcome that of the balanced motions at smaller scales (Rocha et al., 2016b; Qiu et al., 2018, Chereskin et al., 2019), imposing a wavelength boundary beyond which SSH measurements provided by satellite altimetry can no longer be used to infer upper ocean geostrophic flows. Documenting the spatial scale at which this occurs (the so-called transition scale, Lt) for the world ocean has become one of the focal points of recent efforts in the satellite altimetry and SWOT communities (Qiu et al., 2017, 2018; Wang et al., 2018).

Tackling this problem needs high-resolution ocean data, ideally in space and time. To date, progress on documenting Lt has been achieved exclusively through the use of in situ data in a few limited regions and high-resolution global models, given the insufficient time-space resolution of sea surface height (SSH) maps from multi-mission altimetry (these maps have decorrelation scales of ~15 days and 200 km; Chelton et al., 2011; Ballarotta et al 2019). In constructing the altimetric SSH maps, the spatial scales below 200km are severely smoothed by the optimal interpolation algorithm, conserving only a small portion of the signal at small wavelengths (e.g. Ray and Zaron, 2016; Dufau et al., 2016).

Alongtrack altimeter data have a finer spatial resolution than the mapped data, and recent reprocessing now allows us to access oceanic scales down to 50-70 km for Jason–class altimeters, and 35-50 km for Saral/AltiKa (Dufau et al., 2016; Vergara et al., 2019; Lawrence and Callies, 2022). Most of the unbalanced internal tide energy, and some of the internal gravity wave energy, occurs at scales larger than 40 km wavelength and can be observed with the latest alongtrack altimetry data (Zaron, 2019). Using alongtrack SSH data from recent altimetric missions and a statistical approach based on wavenumber spectral analysis, this paper will document the global distribution of Lt. Considering the noise characteristics of different altimetric missions, we limit our Lt estimates to regions where they exceed the local

observability wavelength. We also take into account the uncertainties associated with the altimetric measurements and the influence of this error in our estimates.

Our satellite altimetry wavenumber spectral Lt estimates are consistent with previous studies based on modeling or in-situ analysis: small values of Lt are observed in the highly energetic western boundary current systems and in the vicinity of the ACC (Rocha et al., 2016b, Qiu et al., 2018) suggesting a dominance of geostrophically balanced motions on the surface kinetic energy field. On the other hand, Lt is larger in the vast intertropical ocean (20°S-20°N),

suggesting a significant contribution of energetic wave-type motions to the upper ocean SSH field here.

**2 Data and Methods**

**2.1 Sea Surface Height (SSH) 1Hz data**

Alongtrack SSH data from two missions with different technologies (Jason-3 – J3

conventional nadir altimetry and Sentinel-3A – S3 Synthetic Aperture Radar nadir altimetry) are analyzed at a global scale. The time period analyzed spans their common 4-year period, from March 2015 to March 2019.

Alongtrack SSH observations are maintained at their original 1 Hz observational position with 7 km spacing, and are corrected for all instrumental, environmental, and geophysical

corrections (Taburet et al., 2019). Only time dependent variations of alongtrack SSH measurements are considered, following Stammer (1997), Le Traon et al. (2008) and Xu and Fu (2011, 2012). Since S3 is on a new repeat track, Sea Level Anomalies (SLAs) are computed for both missions by subtracting the Mean Sea Surface model CNES_CLS_2015 (Schaeffer et al. 2016; Pujol et al., 2018) from the alongtrack SSH measurements.

**2.2 Unbiased wavenumber spectrum and spectral shape analysis**

In order to obtain regionally varying spectral estimates, we apply the methodology described in Vergara et al (2019). We sample the alongtrack SSH measurements inside a 12°x12° regional box and then subsample the tracks of each pass inside this regional box to a constant length of 1200 km. Individual spectral estimates are then obtained by performing a spatial Fast Fourier Transform (FFT) on each 1200 km subsample. A Tukey window of 0.5 width is applied to the data in order to minimize boundary effects when performing the FFT over the finite dataset (Tchilibou et al., 2018). Data overlapping is allowed but limited to a 250 km overlap. We verified that the overlapping scale is larger than the local spatial decorrelation scale (estimated from the first zero-crossing of the local autocorrelation function), to avoid an artificial overrepresentation of certain spatial scales introduced by the overlapping. The regional spectrum is then obtained by averaging the individual spectral estimates inside the 12°x12° box. Global coverage is obtained by iteratively repeating this process every 2° in longitude and in latitude.

For each average spectrum, we estimate the 1 Hz error level by fitting a straight-flat line to the SLA Power Spectral Density (PSD) level for wavelengths between 15 and 30 Km wavelength; a similar technique was applied by Xu and Fu (2011); Dufau et al. (2016); Vergara et al (2019). This straight-line fit is horizontal for J2 and S3 (white noise). The spectrum shape of S3 shows a slight slope over the 15 to 30 km wavelength range (red-type noise), which is a characteristic effect of the wind wave field on the SAR measurements (Moreau et al., 2018). The differences on the unbiased spectrum and our methodology when applying either a red noise or white noise fit to the 15-30 km band of S3 data are explored in Appendix A.

The spatial patterns of the noise levels for Jason-3 and Sentinel-3A (Figure 1a, 2a) approximately follow the spatial distribution of significant wave height (Dufau et al., 2016), with peaks in the regions of high sea-state in the North Atlantic, Southern Ocean, and off the

coast of South Africa. The increased SSH noise of the current generation of satellite altimeters due to surface waves is well documented for both radar and SAR altimeters (Tran et al., 2002; Moreau et al., 2018; 2021). The latitudinal trend (Figures 1c, 2c) shows an increase of the noise levels from the equator towards the poles, in agreement with previous studies (Dufau et al., 2016; Vergara et al. 2019). Annual mean Jason-3 wavenumber spectra noise levels range from 1.8 cm rms at the equator to 2.8 cm rms in the Southern Ocean, whereas the Sentinel-3 SAR noise floor is smaller (1.4 cm rms at the equator and 2.3 cm rms in the Southern Ocean). In general, noise levels observed for both satellites indeed show local maxima in the vicinity of the Gulf Stream, Kuroshio extension and the ACC, related to local geophysical effects such as rain cells and more importantly the local wind wave field. Despite the relatively higher noise levels observed in these regions, the mesoscale signal is also strong and therefore the signal to noise ratio remains favorable over these highly energetic regions (Figures 1b and 2b).

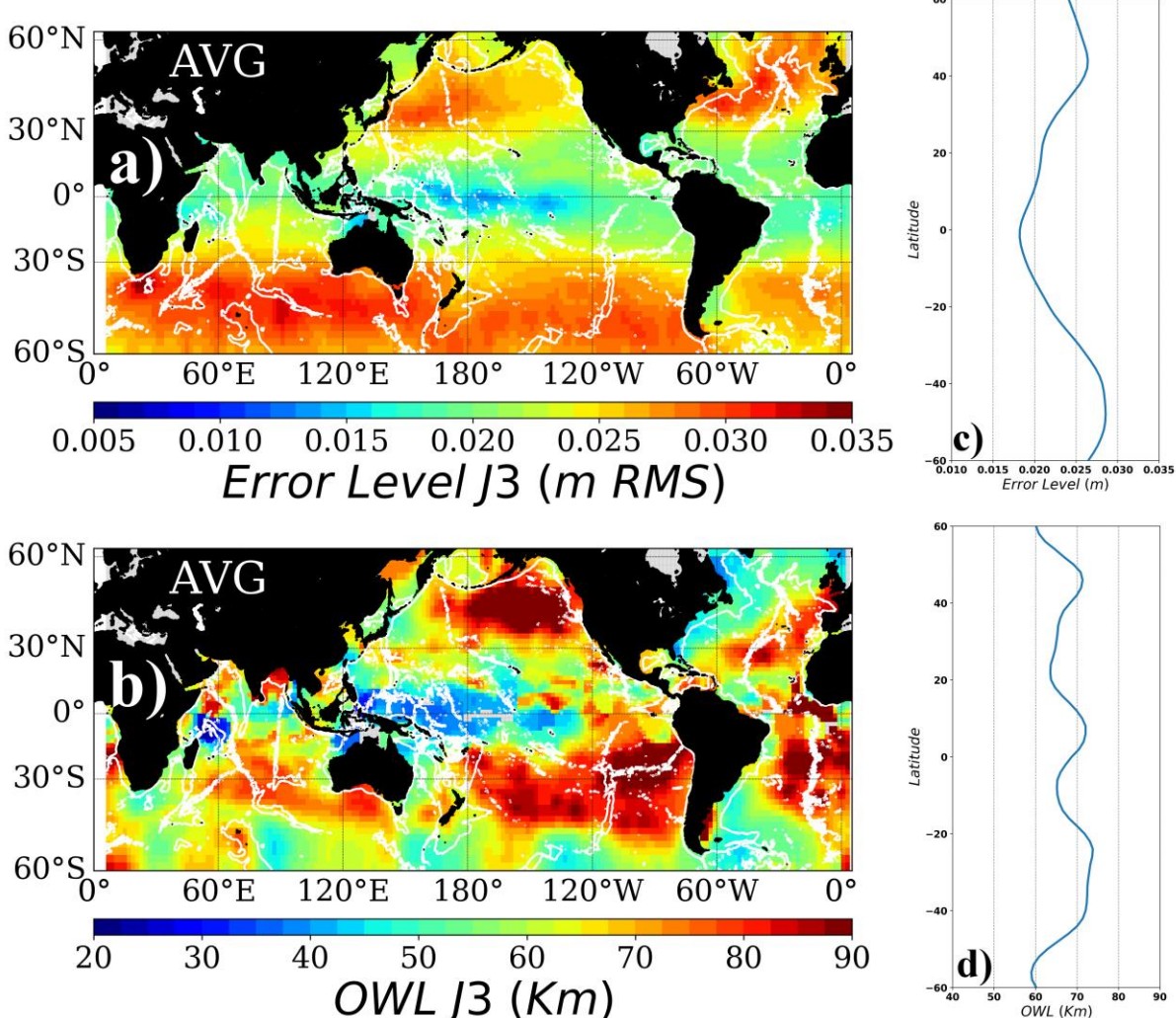

**Figure 1.** (a) Average spectral noise level in meters RMS and (c) its zonal average for Jason-3. Noise level is computed as the average PSD value between 15 and 30 km wavelength. (b) Observable Wavelength, or wavelength for signal-to-noise ratio equal to 1 (in Km) and its zonal average (d). The observable wavelength is computed as the intercept wavelength for the mesoscale spectral slope and the noise level. White contours represent the topography at 3000 m depth.

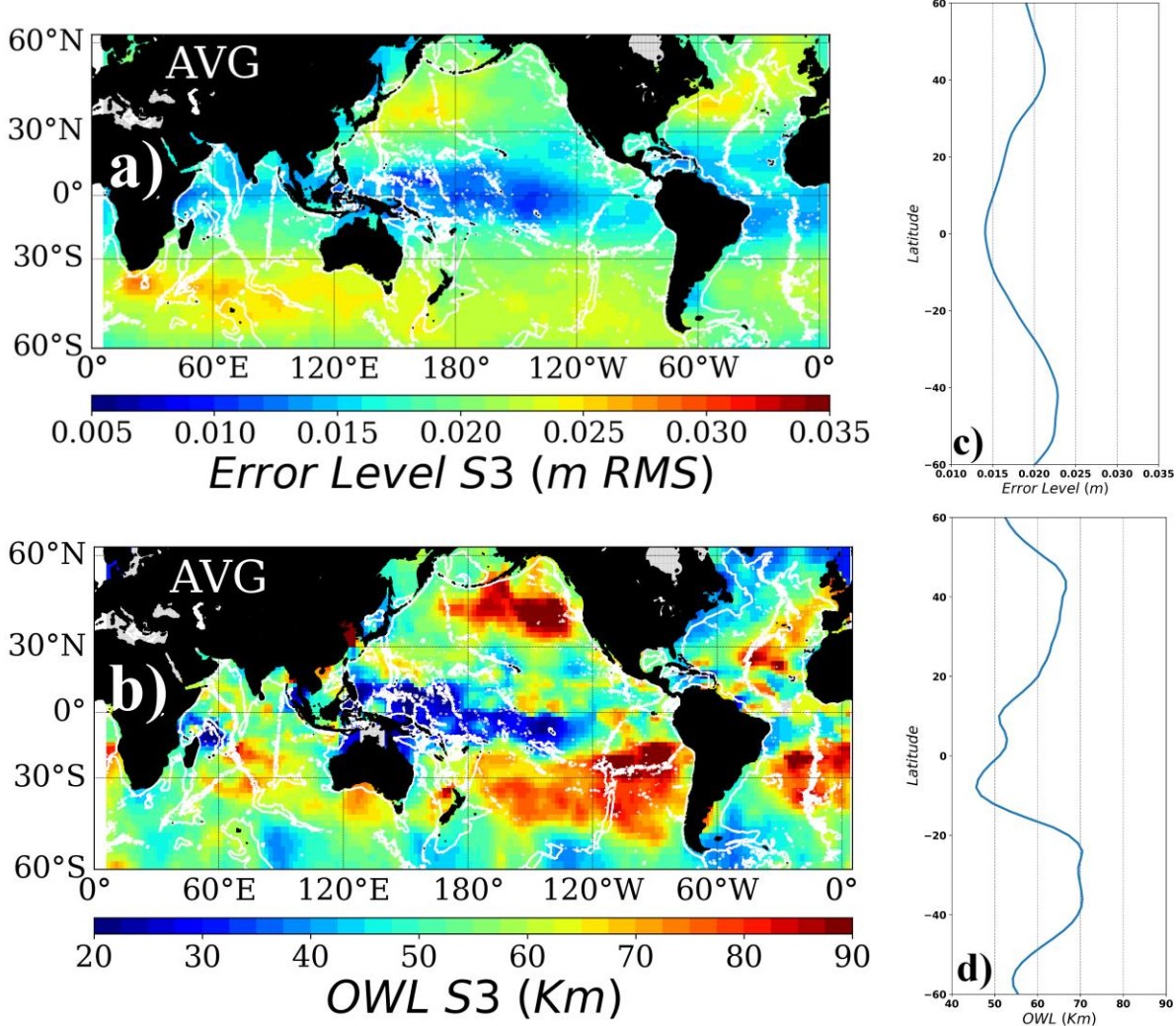

**Figure 2.** Same as Figure 1, but for Sentinel-3A data.

This computed flat noise spectral level is then subtracted from the PSD estimates over the

entire wavenumber range, which provides an unbiased estimate of the regionally-averaged

spectrum (Xu and Fu, 2012). We then analyze the unbiased spectrum in order to determine

two spectral slopes, taking into account the variations of spectral slope values in the fit. The

mesoscale spectral slope is calculated within a geographically variable wavelength range: the

maximum mesoscale wavelength is where the spectral shape significantly (at 95%) departs

from the observed mesoscale spectral slope (usually occurring at wavelengths larger than 500

km), and the minimum regional wavelength limit is based on the local eddy length scale, as in

Vergara et al. (2019). Where possible, a second smaller-scale spectral slope is determined at wavenumbers between 30 km wavelength and the lower mesoscale spectral slope limit.

In order to analyze the two slopes from the regional unbiased spectrum shape, we least-square fit a linear model to the average spectrum obtained from observations in the logarithmic space, defined as:

$$f(x) = {x^{a_1}}/{10^{-b_1}} + {x^{a_2}}/{10^{-b_2}} \qquad (1)$$

where x corresponds to the observed SSH values after applying the Fourier transform, $a_1$ and

165 $a_2$ are the intercepts and $b_1$ and $b_2$ the spectral slopes. This model is therefore defined as the sum of two straight lines in the log-log space, each one representing a different part of the spectrum and capturing a different variability regime. The benefit of performing a simultaneous double-fit for analyzing the spectral shape compared to successive individual least-square fits is two-fold: (1) considering the sum of two linear models preserves the shape

of the observed unbiased spectrum and also allows for curvature where there is a shift in the spectral slope, representing the observed spectrum in a realistic manner. (2) The uncertainty associated with our spectral slope estimates is continuous across the entire wavelength range considered by the model, which is not the case if we consider two successive fits that will minimize the fit errors only for a prescribed wavelength range. We apply this model to each

regionally-averaged unbiased spectrum, between 30 km wavelength and the upper mesoscale wavelength, following Vergara et al. (2019).

The fitting algorithm is initialized using a first guess for the spectral slopes across the wavelength range: for the mesoscale spectral slope we follow Vergara et al. (2019), using the change in spectral slope at low wavenumber and the local eddy length scale as the wavelength

bounds. The small-scale fit is initially computed as a linear fit between the spectral signal at 30 km wavelength and the signal at the local eddy length scale. These values are used for the

first iteration and are recomputed at each step of the least-square minimization procedure to best resolve the double fit, maintaining the end points of the 30 km wavelength and the slope change at low wavenumber boundaries. The minimization process then adjusts the mesoscale and small-scale intercept to better capture the overall shape of the unbiased PSD.

An example of the results of this two-slope methodology are presented in Figure 3a for a region in the north Pacific Ocean, in comparison to the single mesoscale slope fit of Vergara et al. (2019) in Figure 3b. The one-slope mesoscale slope fit in Figure 3b follows the wavenumber curve well within the defined mesoscale range (vertical dashed lines) with a slope of $k^{-4.5}$, and a change in spectral slope is clearly evident at scales smaller than 120 km in wavelength. The two-slope mesoscale fit is slightly steeper in the mesoscale range ($k^{-4.9}$) but the change in slope is well captured at smaller scales down to 30 km in wavelength ($k^{-2.5}$), and the sum of the two linear fits follows the change in curvature of the observed spectral slope (bold solid line). The two-slope linear fit is constrained by inverse-weighting the observations according to the confidence interval of the average spectrum (gray shading in Figure 3a). Using the 4-years of data and multiple tracks within our 12°x12° box, we can expect the linear fit to be well constrained in the mesoscale wavelength range, and the error associated with the estimates of the slope and the intercept to be relatively low. On the other hand, since the confidence interval becomes larger towards smaller wavenumbers (a consequence of subtracting the noise level), the uncertainty in the slope/intercept estimates increases towards shorter wavelengths and the smaller-scale slope fit has higher uncertainty.

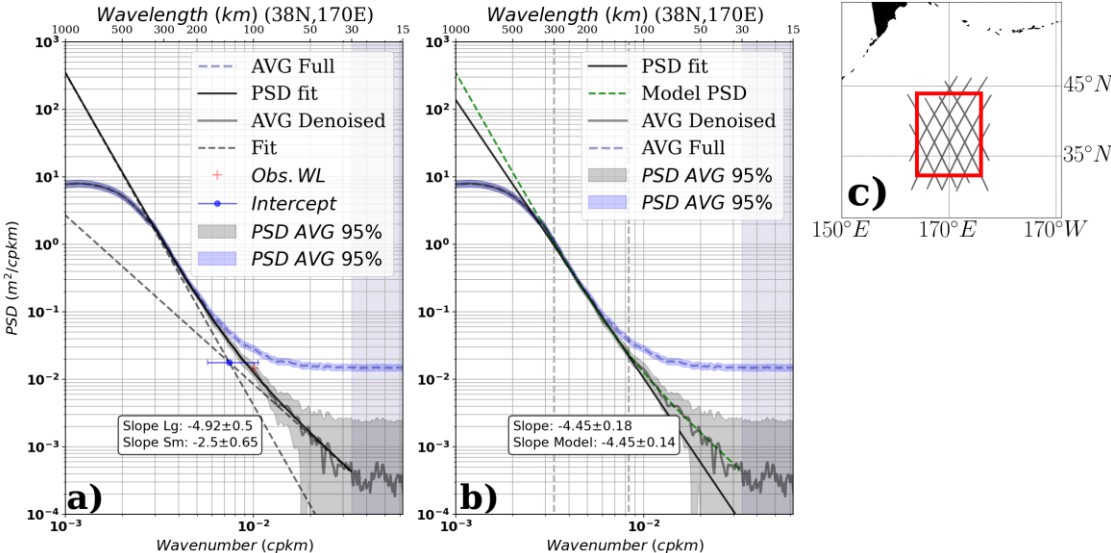

**Figure 3.** (a) Result of the double slope methodology used to characterize the spectral shape over the unbiased power spectral density (PSD) (full gray line). The original and unbiased PSD (dashed blue and gray lines) and the 95% confidence interval of the average PSD (blue and gray shading) are shown. The mesoscale and small-scale spectral slope fits (dashed black), combined double linear fit model (full black), and the Lt intercept of large- and small-scale spectral slopes (blue dot, including its uncertainty), are also illustrated. (b) Single mesoscale fit method, as in Vergara et al. (2019). As a reference, the result of the PSD fit from 3a is also plotted (dashed green line). Unbiased average PSD (full gray line) and 95% confidence on the average estimate (gray shading) analyzed using the wavelength range depicted by the vertical dashed lines to characterize the spectral slope. Average spectral slope and their corresponding 1-sigma values are indicated in the inserts. (c) Zone where the average PSD sample is computed from J3 data over 2015-2019.

### 2.3 Observability wavelength

The Observability Wavelength (OWL) is defined as the threshold wavelength where the SSH spectral signal exceeds this flat noise level (i.e. SNR > 1). Given that the first Rossby radius of deformation and the eddy length scales (Eden, 2007) both generally decrease towards higher latitudes whereas the noise level increases due to higher sea-state, one would expect

that the OWL scales would increase towards high latitudes. The zonal average of the Observable Wavelength (OWL) for both satellites is summarized in Figures 1d and 2d. The combined regional variability of the mesoscale spectral slope and the noise levels both contribute to the complex observed patterns of the OWL (Figures 1b, 2b). For regions with strong mesoscale variability signals (e.g. Southern Ocean, Gulf Stream, Kuroshio, Agulhas current), the local observable wavelength is short despite relatively high noise levels. The observable wavelength for Jason-3 varies from 40 km in the western tropical Pacific, 50-60 km in the western boundary currents and can reach 90 km in the low energy Eastern North Pacific due to the higher noise levels. Zonal averages across these regional patterns lead to values between 60-70 km (Figure 1d), whereas the zonally averaged OWL for Sentinel-3 reaches 65-70 km in the mid latitudes, but only 50 km in the equatorial band.

**2.4 Uncertainty analysis for the intercept wavelength**

In addition to the fitting parameters for the model described by Eq. (1), we compute the uncertainty associated with the least-square fitting, related to each parameter. This helps us in the interpretation of the results by allowing us to estimate the validity of the spectral slope values for the large and small wavelength ranges, and also their intercept.

The uncertainty (or error) emerges from the confidence interval envelope obtained when computing the regional average spectrum (gray shaded area in Fig. 3a and 3b). On this log scale, the 95% envelope of the average PSD appears to grow considerably towards high wavenumbers. This is a consequence of the denoising method: the 95% envelope is impacted by the subtraction of the noise plateau computed between 15 and 30 km wavelength and the effect will become more evident in the high wavenumber part of the spectrum, given that the smaller amplitude of the PSD values is closer to the noise level.

Using the uncertainty estimates from the mesoscale and small-scale spectral slopes, we can

determine the error associated with their intercept by propagating the uncertainty as:

$$\delta L_t = L_t \cdot \left( \frac{\delta b_1 + \delta b_2}{b_1 + b_2} + \frac{\delta a_1 + \delta a_2}{a_1 + a_2} \right) \tag{2}$$

where $L_t$ is the intercept wavelength, $a_1$ and $a_2$ the slopes and $b_1$ and $b_2$ the intercepts of the

two lines fitted to the observed PSD. The 1-σ deviations from the fitted parameters a and b are

denoted by δ.

## 3. Results

In this section, we analyze the temporal mean geographical distribution of the mesoscale and

small-scale spectral slopes computed using the model described by Eq. (1). Additionally, we

estimate (when possible) the intercept wavelength between the two slopes in the spectral

space. This characteristic intercept wavelength may be considered as a first-order approach to

the transition scale ($L_t$) calculated from model analyses by Qiu et al (2018), and from *in situ*

observations in Qiu et al (2017). However, we do not analytically separate the contributions

of the first-three baroclinic modes of internal gravity waves (IGW) as in Qiu et al (2018), so

the intercept scale that we present here is a statistical position from SSH wavenumber spectral

slope changes, and not a dynamical calculation. The spectral slope changes can delineate the

separation of mesoscale balanced motions from a combination of small-scale balanced

motions, unbalanced motions and altimetric observation errors.

### 3.1 Meso- and small-scale spectral slopes

The methodology used to analyze the spectral shape does not explicitly separate the

geostrophically balanced mesoscale regimes from small-scale regions of the spectrum.

However, it allows us to infer their associated contributions to the observed PSD by means of

the change in spectral slopes and their intercept. We will refer to the results of either part of the bi-linear model as meso- and small-scale spectral slopes.

### 3.1.1 Mesoscale spectral slope

The geographical distribution of the mesoscale spectral slope values is very close for the two missions analyzed (Figures 4a and 5a), showing larger slope values over the Western boundary current systems, as well as the Antarctic Circumpolar Current, indicative of the energetic mesoscale circulation that is observed in both regions. The observed spectral slope values in these highly-energetic regions vary between $k^{-4}$ and $k^{-5}$, for both Jason-3 and

Sentinel-3A, and are in general agreement with the distribution reported in Vergara et al. (2019). Conversely, the lowest spectral slope values are observed in the intertropical band (equatorward of 15°), with slope values between $k^{-2}$ and $k^{-3}$, also consistent with values obtained using a simple fit methodology (Xu & Fu, 2012; Dufau et al., 2016; Vergara et al., 2019).

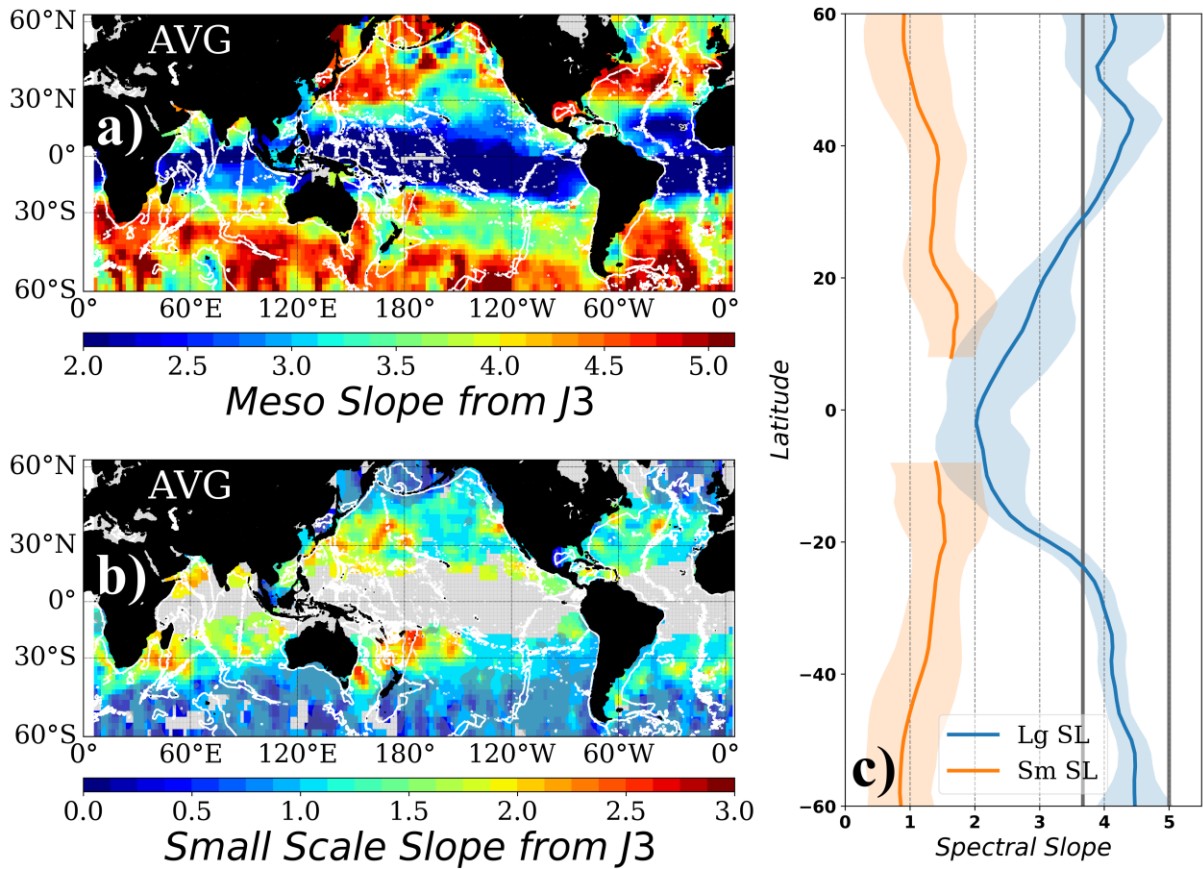

**Figure 4.** (a) Mesoscale and (b) small-scale spectral slopes fitted to the observed unbiased PSD from Jason-3. (c) Zonal average of the mesoscale (Lg SL) and small-scale (Sm SL) spectral slopes in (a) and (b). Color shading around the average values correspond to the 95% confidence interval for the zonal mean. Vertical gray lines denote the $k^{-11/3}$ and $k^{-5}$ spectral slope values. Gray shading in (b) corresponds to zones where the uncertainty associated with the slope estimate is higher than 40% of the slope value. Blanks zones in (b) corresponds to zones where the double slope model does not describe the observed shape of the spectrum. White contours represent the topography at 3000 m depth.

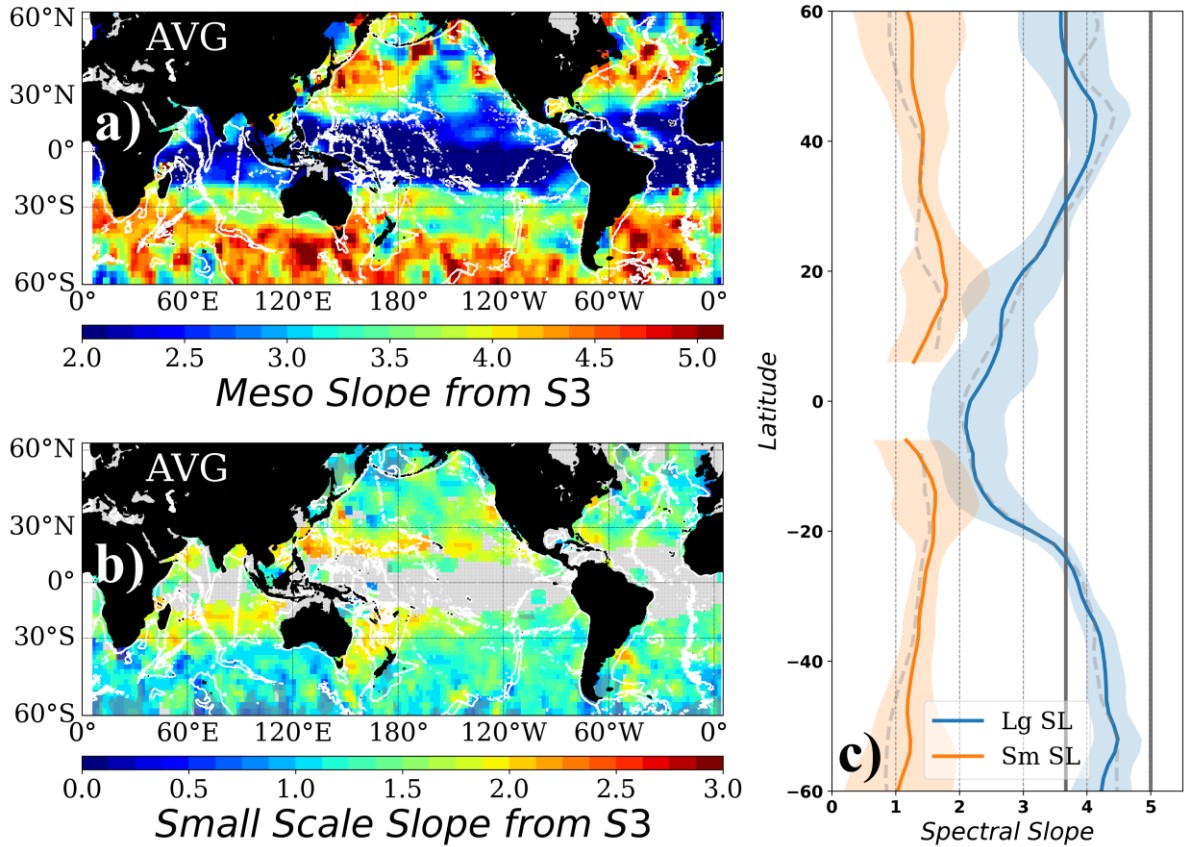

**Figure 5.** Same as Figure 4 for Sentinel-3: (a) Mesoscale and (b) small-scale spectral slopes fitted to the observed unbiased PSD from Sentinel-3A. (c) Zonal average of the mesoscale (Lg SL) and small-scale (Sm SL) spectral slopes in (a) and (b). Color shading around the average values correspond to the 95% confidence interval for the zonal mean. Vertical gray lines denote the $k^{-11/3}$ and $k^{-5}$ spectral slope values. Gray shading in (b) corresponds to zones where the uncertainty associated with the slope estimate is higher than 40% of the slope value. Blank areas in (b) correspond to zones where the double slope model does not describe the observed shape of the spectrum. Dashed gray lines in (c) correspond to Jason-3 zonal averages from Figure 4c. White contours represent the topography at 3000 m depth.

In some regions, the mesoscale spectral slope values obtained are slightly lower (flatter slopes) for Sentinel-3A than for Jason-3. This could be related to the white noise level model

used for S3A. Red noise is often observed in the 1Hz SAR data at small wavelengths related to wave and swell impacts on the signal processing (Moreau et al., 2018). Our methodology of removing a white noise level computed over the 15-30 km wavelength range could impact the unbiased PSD up to ~150 km wavelength, leaving a remanent of energy associated with the SAR processing of ocean swell effects (Moreau et al., 2021) over the mesoscale and small-scale range that will act to flatten the spectral slope at the mesoscale wavelength range. We tested the differences in the spectral PSD levels related to fitting a red or white noise model for S3A in Appendix A, illustrated in Figure A1. In general, the difference in the unbiased curves appears at high wavenumbers having a smaller PSD amplitude, but the differences in the mesoscale spectral slope are generally small.

Indeed, the zonal annual-mean distribution of the mesoscale spectral slope values is the same for both missions (Figure 4c and 5c, in blue), showing a profile that is nearly symmetrical around the equator, with values increasing poleward, ranging between $k^{-11/3}$ and $k^{-5}$ for latitudes poleward of 20°S/30°N and decreasing sharply towards $k^{-2}$ around the equator. These values confirm that the mid-latitude SSH mesoscale spectral slopes vary within the regimes of sQG to QG dynamics, whereas the tropical band has much flatter spectral slopes that reach $k^{-2}$, in agreement with previous altimetric studies (Xu and Fu, 2012; Vergara et. al.; 2019).

### 3.1.2 Small-scale spectral slope

The originality of this method is to estimate the second small-scale slope from the wavenumber spectra when possible. The wavelength range for this small-scale slope varies geographically, but is generally between 30 to ~150 km wavelength. This region of the spectrum is where we expect to observe the upper ocean dynamics to shift from a regime dominated by geostrophically balanced motions (eddy-like mesoscale structures in an SQG or

QG regime) towards a wave-like regime where the SSH variability is dominated by unbalanced motions (i.e., IGWs, coherent internal tides and the cascade of energy from non-phase locked internal tides). For the current generation of satellite altimetry observations,

these 30-150 km wavelengths of the SSH spectrum can also be influenced by residual altimetric geophysical errors and may approach the observability capabilities of each instrument (OWL). Therefore, the results on the small-scale SSH spectral slope estimates need to be interpreted in light of their inherent limitations, taking into account the uncertainty in the slope determination, residual altimetric errors, and the local observable wavelength.

On average, the linearly fitted small-scale spectral slopes vary between $k^{-1}$ and $k^{-2}$ for both S3A and J3 (Figures 4b and 5b), with slightly higher slope values over the western part of each ocean basin (around $k^{-2}$ at 30° of latitude) in mid-latitudes compared to the eastern basins. The meridional distribution of valid slope values shows a decrease of the small-scale spectral slope values towards the poles, as well as an increase in the dispersion around the

average values (i.e. the uncertainty envelope tends to grow as latitude increases). The zones where the uncertainty associated with the small-scale slope estimate is higher than 40% of the slope value are shaded in grey. This value was chosen in order to discard estimates with high uncertainty associated with the bilinear fit from the analysis. These regions are generally at higher latitude, and with slopes close to, or less than, $k^{-1}$. As latitude increases, the first

baroclinic Rossby radius decreases and therefore the mesoscale slope fit will be made over progressively smaller wavelengths. This pushes the small-scale fit to be estimated over a very narrow wavelength range above the 30 km noise cutoff, having large uncertainties.

Note that the optimal fitting algorithm is not able to separate the contribution of two different spectral slopes in the intertropical band equatorward of ~20°. In this zone, we observe that the

SSH signal is essentially captured by the flatter meso-scale spectral slope of $k^{-2}$ (Figures 4c, 5c), extending to small wavelengths, and although the least-squares double slope fit does

estimate a small-scale spectral slope, it is not significant (reduced by several orders of magnitude in terms of energy content compared to the mesoscale slope contribution). These cases have been left blank in Figures 4b and 5b.

355 The zonal-mean values of the valid small-scale slopes (Figs 4c and 5c) show a $k^{-1.5}$ spectral slope between 10-20° latitude for both J3 and S3A. These values come from isolated patches in the western Pacific, the Indian Ocean, and in the zone of tropical instability waves (up to ~10°N) in the north-eastern Pacific. In the mid-latitudes, from 20-45° in latitude, the small-scale slope decreases from $k^{-1.5}$ to around $k^{-1.3}$ for both missions, with higher values near $k^{-2}$ in

360 the western basins. Any differences in the small-scale spectral slope for S3A, related to the removal of the white spectral noise, are small, and within the small-scale error range in Figs 4c and 5c.

**3.2 Intercept wavelength**

The intercept of the fitted meso- and small-scale spectral slopes results can be used to obtain a

365 characteristic wavelength for the change in their dynamical regime. If we consider that the mesoscale slopes reflect the balanced (s)QG dynamics at mid latitudes, and the valid small-scale slope values reflect the wave-like motions from internal tides or IGWs, the intercept wavelength therefore indicates the boundary at which the SSH variability would be mainly driven by either dynamical regime. This wavelength scale could be considered as an

370 approximation of the so-called transition scale from balanced to unbalanced motions (Qiu et al. 2017; 2018), which indicates the boundary between the circulation dominated by either geostrophically balanced or unbalanced motions (in terms of SSH variability). While Qiu et al. (2018) calculate the transition scale by explicitly filtering the contribution of balanced/unbalanced motions from the PSD in the wavenumber/frequency space, we compute

the spatial scale directly from the observed PSD, assuming that its shape captures both

dynamical regimes.

Considering the limitations inherent to satellite altimetry observations (e.g. noise level, residual errors from corrections and observability wavelength), we also compute the uncertainty associated with our estimates for the intercept wavelength. We then exclude any

results that have the following criteria: (1) uncertainty in the mesoscale and/or small-scale spectral slope higher than 40%, (2) intercept wavelength value being less than the local observable wavelength (OWL). Following these criteria, the intercept wavelengths we can interpret from alongtrack altimetry are reduced to a fraction of the world ocean. Nevertheless, within these constraints, intercept wavelength spatial distribution shows higher values in the

tropical band and lower values towards the poles for the two missions considered (Fig. 6). Both J3 and S3A show intercept wavelengths around 100 km on average in mid latitudes from 25-45°, reaching to 140-160 km in the tropics near 10° latitude (Fig 6c and 6d).

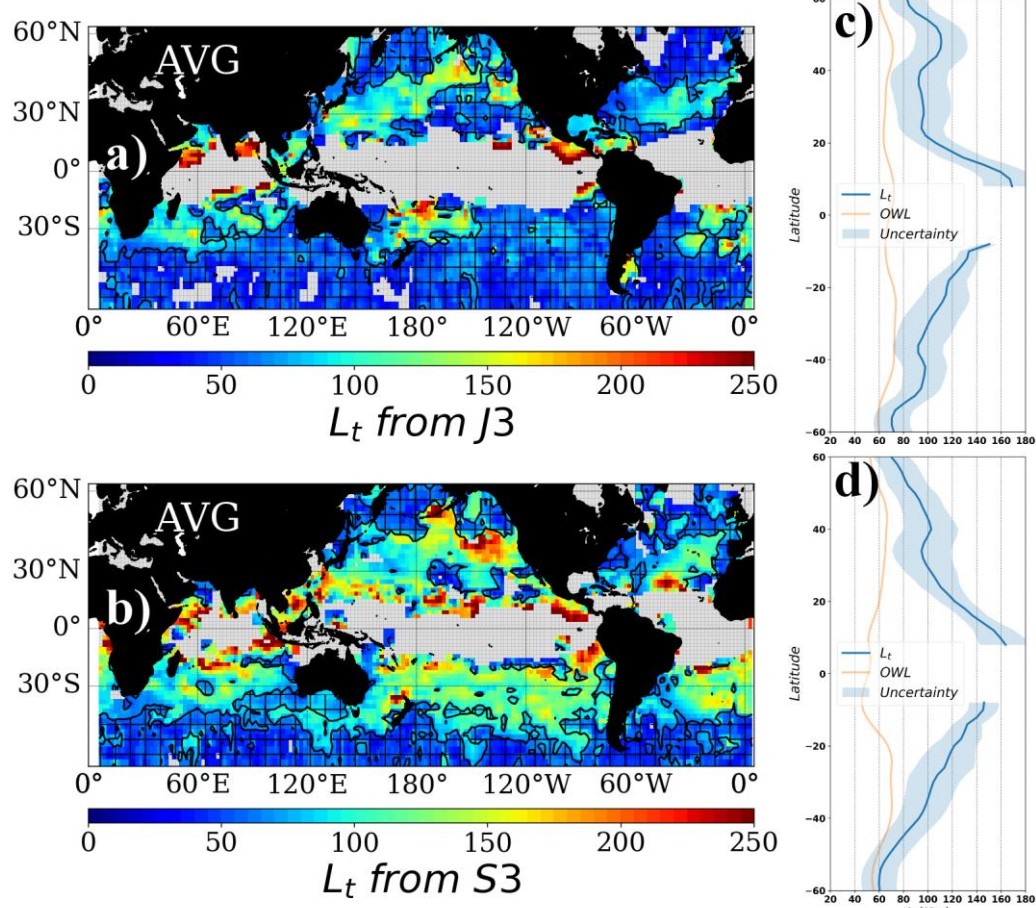

**Figure 6.** Intercept scale (in Km) between the large- and small-scale spectral slopes for J3 (a) and S3 (b). Shaded/hatched areas correspond to regions where the intercept scale is smaller than the local observable wavelength (signal-to-noise ratio equals 1). Blank areas in (a) and (b) correspond to the regions where the observed PSD is accounted for using a single slope approach. (c, d) Zonal averages of (a) and (b). Uncertainty envelope is also included in the zonal average. Orange lines in panels (c) and (d) correspond to the zonal average of the observable wavelength for J3 and S3A respectively (Figures 1d and 2d). Note that the average intercept scale is always higher than the average OWL. We also verified that the first baroclinic Rossby radius of deformation is larger than the computed intercept values. White contours represent the topography at 3000 m depth.

Using a state-of-the-art global ocean simulation, Qiu et al. (2018) recently explored the geographical and seasonal variations of the balanced to unbalanced transition scale, highlighting that the highly energetic western boundary current systems have relatively short

transition scales, with the largest transition scale values occurring in the relatively low-energy regions bounded by the intertropical and subpolar bands. Their modelled regional distribution reflected the local levels of mesoscale variability and the energy levels of unbalanced motions (near-inertial flows, internal tides and inertia-gravity waves). Large transition scale values are also observed where prominent bathymetric features exist (Qiu et al., 2018) such as the North Atlantic Ridge, and the Western Equatorial Pacific.

The observed J3 and S3A intercept values across each basin are similar to recent modelling results of global estimates for the balanced/unbalanced motions transition scale, with shorter transition wavelengths located in the energetic western boundary current regions and longer values in the eastern basins. Using the points that can be defined from our estimates, the Kuroshio Extension has intercept values of around 90-100 km for both J3 and S3, the Gulf Stream having values of around 60 km (Figure 7). Whereas the eastern North Pacific intercept wavelengths reach values of 120-140 km. We note that S3-A with lower noise has more geographical coverage of Lt estimates within each defined box, leading to local differences in the average meridional Lt distribution. Observations in the Antarctic Circumpolar Current are unfortunately non-interpretable. The sharp mesoscale spectral slopes observed here (related to the highly energetic local eddy activity at small Rossby radii), result in an intercept wavelength scale around 50 km or less. Given the higher noise levels, from higher sea-state in the Southern Ocean, these intercept scales are below the local observable wavelength. Observational results point out that the balanced/unbalanced transition scale for this region often shows values around 30 km (Drake Passage region; Rocha et al.; 2016a), which is currently inaccessible with satellite altimetry observations.

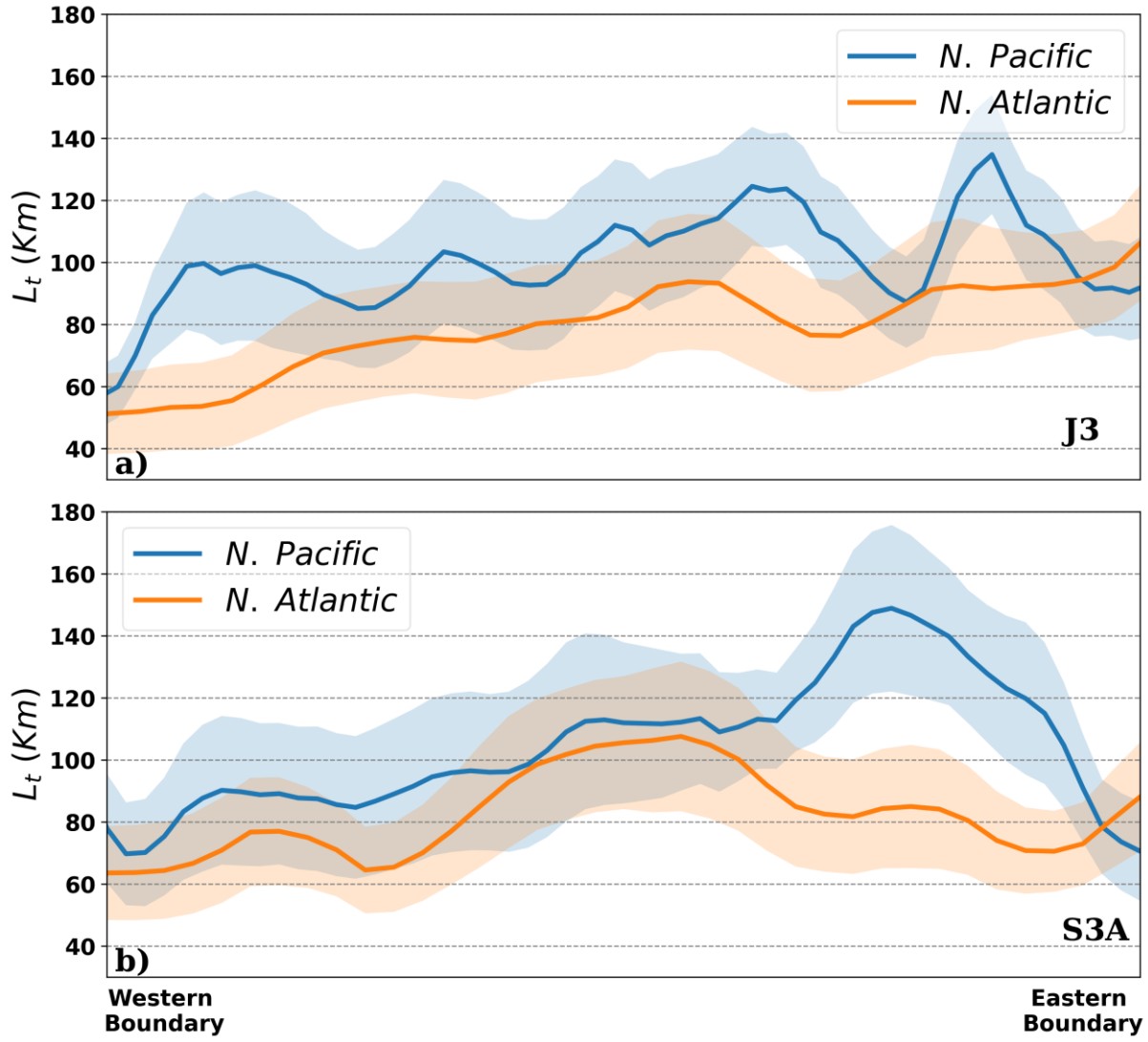

**Figure 7.** Meridional average of the intercept wavelength presented in Figure 4 over the North Pacific (130°E,30°N-240°E,50°N) and North Atlantic (280°E,30°N-350°E,50°N), for Jason-3 (a) and Sentinel-3A (b). The envelope around the averages indicates the 1-sigma deviation.

Equatorward of 25°, the intercept scale shows the highest values often exceeding 150 km, which suggests that the dynamical regime characterized by shallow spectral slopes is the main contributor to the observed SSH variability. Modelling results show that the kinetic energy of the unbalanced motions dominates the SSH variability at the low latitude regions, with transition scale values exceeding 200 km (Qiu et al., 2018).

## 4. Summary and discussion

We have in this study explored the capability of currently available alongtrack data to capture the changes in the circulation variability at wavelengths shorter than 150 km. We used a statistical approach consisting of analyzing the shape of the SSH power spectral density, which can be indicative of the underlying circulation dynamics. In addition to the mesoscale spectral slope, we compute a secondary spectral slope at smaller spatial scales, in a wavelength region that is characterized by a regime change from geostrophically balanced mesoscale motions to a potentially non-geostrophic regime. The methodology used here is based on an unbiased slope estimate, after removal of a white-type instrument noise. However, the least-squares fit takes into account the variance of the errors associated with the instrumental noise, which grows towards the high-wavenumber part of the SSH spectrum as the signal amplitude decreases, and therefore increases the uncertainty of the estimated parameters towards short wavelengths.

A second outcome was to compute the intercept of the meso- and small-scale spectral slopes estimated in order to obtain a characteristic transition wavelength. We interpret this wavelength as a proxy for the transition scale that marks the boundary between the geostrophically balanced and unbalanced motions in the SSH signal. Despite the limited number of intercept wavelengths that pass our rigorous selection criteria, these intercept wavelengths show a distinctive geographical pattern. Higher spectral slope transition values are located around the tropics, sometimes exceeding 200 km in wavelength, and towards the eastern ocean boundaries (between 100-150 km), in agreement with circulation patterns with important wave-like variability at the mesoscale wavelength range (Pollmann, 2020; Tchilibou et al., 2018) compared to the local eddy field. The shortest intercept wavelengths are on the other hand observed over the western boundaries and towards high latitudes, where the circulation variability is dominated by an energetic mesoscale eddy field.

## 4.1 Uncertainty in the mesoscale spectral slope

In our 2-slope methodology, the larger mesoscale spectral slope estimate starts from a first guess based on the geographically variable wavelength range specified in Vergara et al (2019). Then, the least-squares minimization of the 2-slope fit allows this minimum wavelength range to be adjusted. As opposed to previous studies, we also include in our spectral slope estimates the inherent uncertainty that is contained in the altimetric observations. We have compared the results of the mesoscale spectral slopes from the bilinear solution against the observed spectrum (Fig. 8). For consistency, we used the locally variable wavelength range proposed in Vergara et al. (2019) to compute the mesoscale spectral slopes. Overall, we observe that the differences in spectral slopes when diagnosing either the optimal fit solution or the observed spectra are small, of order 0.1 across the world ocean. The differences are slightly higher in the equatorial regions (from both datasets) between 0.3 to 0.5 (Fig. 8), but remain smaller than the average uncertainty associated with the mesoscale spectral slope for the bilinear fit for these latitudes (Figures 4 and 5).

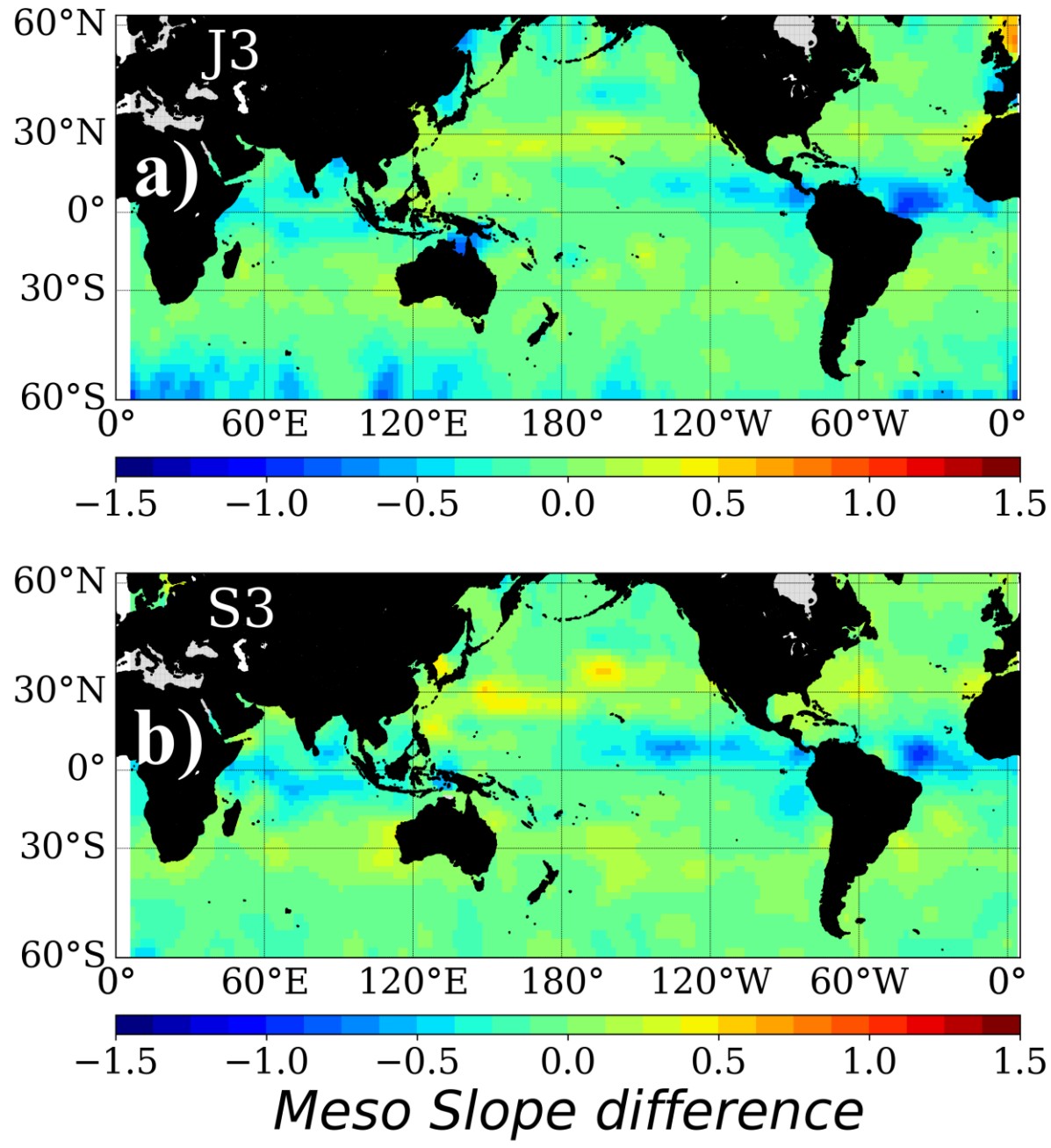

**Figure 8.** Difference in the mesoscale spectral slope estimates computed over the fitted solution (from eq. 1) and the observed spectrum using the simple linear fit approach of Vergara et al. (2019). (a) Jason-3 and (b) Sentinel-3A.

## 4.2 Uncertainty in the small-scale wavelength range

At wavelengths shorter than 150 km, the analysis of the SSH spectral shape becomes increasingly sensitive to the observation errors (i.e. instrumental error, accuracy of the altimetric corrections) and therefore the interpretation of the results at high wavenumber need to account for the increased uncertainty compared to the mesoscale wavelength range. Our methodology for the analysis of the unbiased spectral shape assumes a white noise plateau for the J3 and S3A 1Hz observations which is removed to reveal the SSH spectrum free of instrumental errors. Using this first-order approximation for the instrumental error significantly increases the uncertainty of the spectral estimates towards high wavenumbers in comparison to their weak amplitude (i.e. the 95% CI envelope grows as we move towards the small-scale part of the spectrum). Thus, the uncertainty in the estimates compared to the signal is more important in the small-scale part of the spectrum than in the mesoscale wavelength range. This uncertainty also affects the estimates of the intercept wavelength in equation (2).

In addition, in regions where the first baroclinic Rossby radius is small (e.g. high latitudes) and/or the mesoscale energy is intense, the mesoscale spectral slope dominates the double fit and extend down to small wavelengths. In this case, there is not much wavelength range above 30 km to perform a second-slope fit, and this combines with the larger error variance at small scales. We therefore observe an inverse relationship between the error associated with the small-scale spectral slope and the wavelength range used to perform the small-scale fit (Fig. 9) (i.e. the range between the intercept wavelength and the 30 km wavelength, the limit for computing the noise plateau). We note that Jason-3 (Fig 9a) has higher error variance than S3A (Fig 9b), as expected, and this larger error extends over a longer wavelength range. Whereas the S3A small-scale spectral slope to error ratio tend to be confined to smaller wavelength ranges. Note that values of small-scale slope error ratio > 0.4 were discarded from the analysis of intercept wavelengths as estimates having high uncertainty, and Figure 9

explains why more regions are eliminated with this 0.4 cutoff for Jason-3 analyses, than for S3A (see Figures 4b, 5b and 6). We also discarded all the intercept wavelength estimates that were below the SNR=1 level, which delimits the observable wavelength in altimetric observations. The small intercept values observed at high latitudes (often smaller than 50 km wavelength) were therefore classified as non-interpretable, even though their distribution agrees with the results of Qiu et al (2018) for the SSH-based transition scale estimates, in particular the short Lt values observed around the Antarctic Circumpolar region.

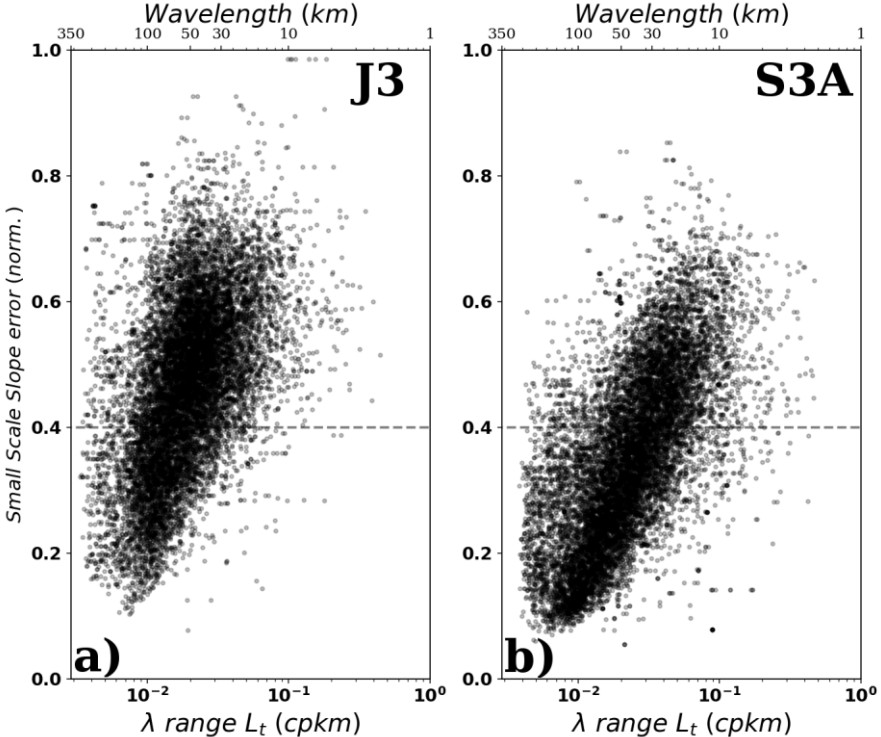

**Figure 9.** Error in the determination of the small-scale spectral slope plotted against the wavelength range between the computed intercept scale and 30 km wavelength (noise level limit) for Jason-3 (a) and Sentinel-3A (b).

At these smaller spatial scales, the observed variability may result from different sources, both geophysical and instrument/platform related, and the diagnosed small-scale spectral slope is potentially a combination of such elements. Among the dynamical contributions, it

has been shown that a significant part of the SSH PSD spectrum at wavelengths shorter than 150 km is related to phase-locked and non phase-locked internal tides (Ray and Zaron, 2016). An important cascade of energy is apparent in the SSH spectrum around the tropical latitudes (Tchilibou et al., 2018), with an increased high-frequency variability of tidal origin (mainly non-phase locked) for wavelengths shorter than 70 km (Tchilibou et al. 2022). Using a high-resolution global OGCM, Qiu et al. (2018) explored the influence of the internal tides on their estimates of Lt, concluding that the removal of the phase-locked internal tide contribution significantly reduces the values of Lt by 50 to 100 km in many regions. Today, only a few phase-locked tidal constituents are available as a potential altimetric correction (e.g. M2; Zaron, 2019). So a significant portion of the SSH PSD in our analysis may be influenced by internal tide variability.

In addition to the dynamical contributions, the altimetry-based small-scale spectral slope estimates may also be influenced by errors in the altimetric measurements used. One source of error that has been characterized at wavelengths ranging from 30 to ~80 km are the imprecisions related to the Mean Sea Surface Model (MSS) used to compute the altimetric Sea Level Anomalies (SLA), which have been quantified to contribute as much as 30% of the observed SLA variability (Pujol et al., 2018). The benefit of the latest MSS model, CNES_CLS_2015, is a reduction of the associated error by at least half compared to conventional models (Pujol et al., 2018). Nevertheless, this could still be a source of errors at short wavelengths for recent uncharted missions such as S3A. We performed sensitivity tests on the impact of the MSS model on our estimates at small-scale (not shown), revealing SSH PSD may increase but the spectral shape is preserved, and therefore the estimates of the small-scale spectral slope do not significantly change. This effect is comparable to the noise plateau differences presented in Appendix A for S3A.

**4.3 Seasonality of the spectral shape and intercept wavelength**

The seasonal variability of the spectral characteristics derived from altimetric observations has been documented in recent literature (e.g. Dufau, et al., 2016; Vergara et al., 2019; Lawrence and Callies; 2022), highlighting the fact that the variations of the spectral shape are related to changes in both the underlying circulation and surface ocean stratification as expected, but also to variations of the altimetric noise levels throughout the year. The methodology used in the present paper also reveals a marked seasonal change of the spectral slopes, with variations in the mesoscale wavelength range showing sharper values during summer months than during winter, as a byproduct of the interaction between the higher noise levels during winter and the presence of small-scale turbulence that is generated through vigorous vertical mixing (Sasaki et al., 2014; Callies et al., 2015). This small-scale variability is therefore partially masked by the increased noise levels (and increased uncertainty in our slope estimates) during winter months. On the other hand, during summer months the instrumental noise levels drop, hence the SSH observability spans a large wavelength range with favourable signal-to-noise ratio. During summer months, higher spectral slope values are consistent with interior QG dynamics, suggesting that large eddies are formed through baroclinic instabilities in the thermocline and very little energy cascades to smaller scales (e.g. Callies, et al., 2015). The combination of favourable conditions for the generation of eddies at mesoscales (larger than 100 km) and lower noise levels provide an ideal altimetric observability scenario during summer months. The seasonal variability of the meso- and small-scale spectral slopes is documented in Appendix B.

The intercept wavelength also modulates seasonally, suggesting that information about the SSH variability in the sub-100km wavelength range is effectively reflected by this parameter computed from along-track altimetry observations (Figures 10 and 11). Changes in the upper ocean stratification will significantly modulate the energy levels of balanced and unbalanced motions, that collectively contribute to the SSH variability observed for wavelengths ranging

from 15 to 200 km. During summer months, a shallow mixed layer with a sharp density gradient at its base works to enhance the surface unbalanced motion kinetic energy (Rocha et al., 2016b), that surpass the energy levels of the geostrophic turbulence at <100 km wavelength range. Conversely, the vigorous vertical mixing observed during winter energizes

575 the balanced motions in the small-scale part of the spectrum, leading to a predominance over unbalanced motions in the SSH variability during this season (Rocha et al., 2016b; Qiu et al., 2017). The modeling results reported by Qiu et al. (2018) show that this pattern is nearly ubiquitous in the global ocean, with larger transition scale values observed during summer than winter. Our seasonal estimates of the intercept scale show a similar pattern both in terms

of eastern-western basin asymmetry and larger transition scales in summer in the zonal-averages (Figures 10 and 11; Appendix B).

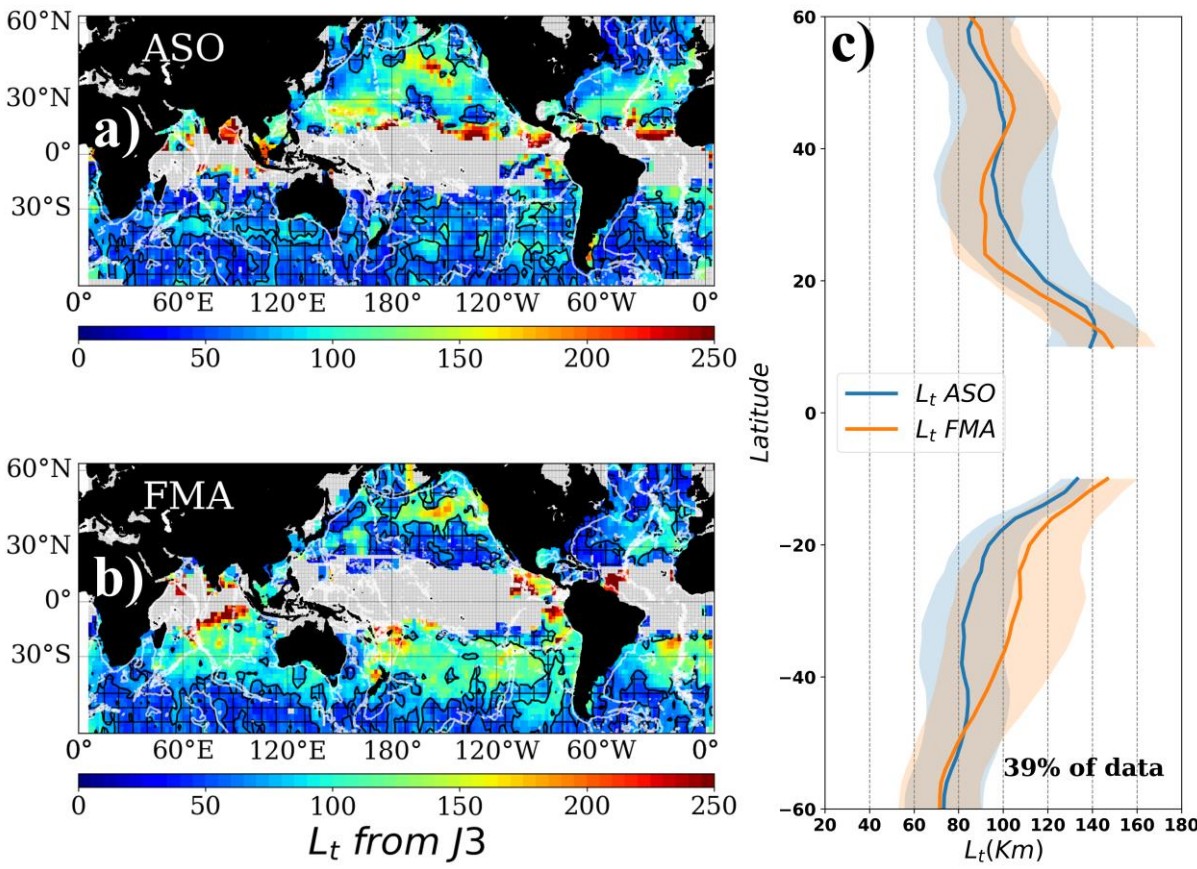

**Figure 10.** Intercept scale (in Km) averaged during (a) August-September-October and (b) February-March-April for Jason-3. Shaded/hatched areas correspond to regions where the

intercept scale is smaller than the local observable wavelength (signal-to-noise ratio equals 1). Blank areas in (a) and (b) correspond to the regions where the observed PSD is accounted for using a single slope approach. (c) Zonal averages of the non-masked areas of (a) and (b), but only considering the pixels where the observations of (a) and (b) are different at 95% confidence. The percentage of data that meets this criterion is also indicated. White contours in (a) and (b) represent the topography at 3000 m depth.

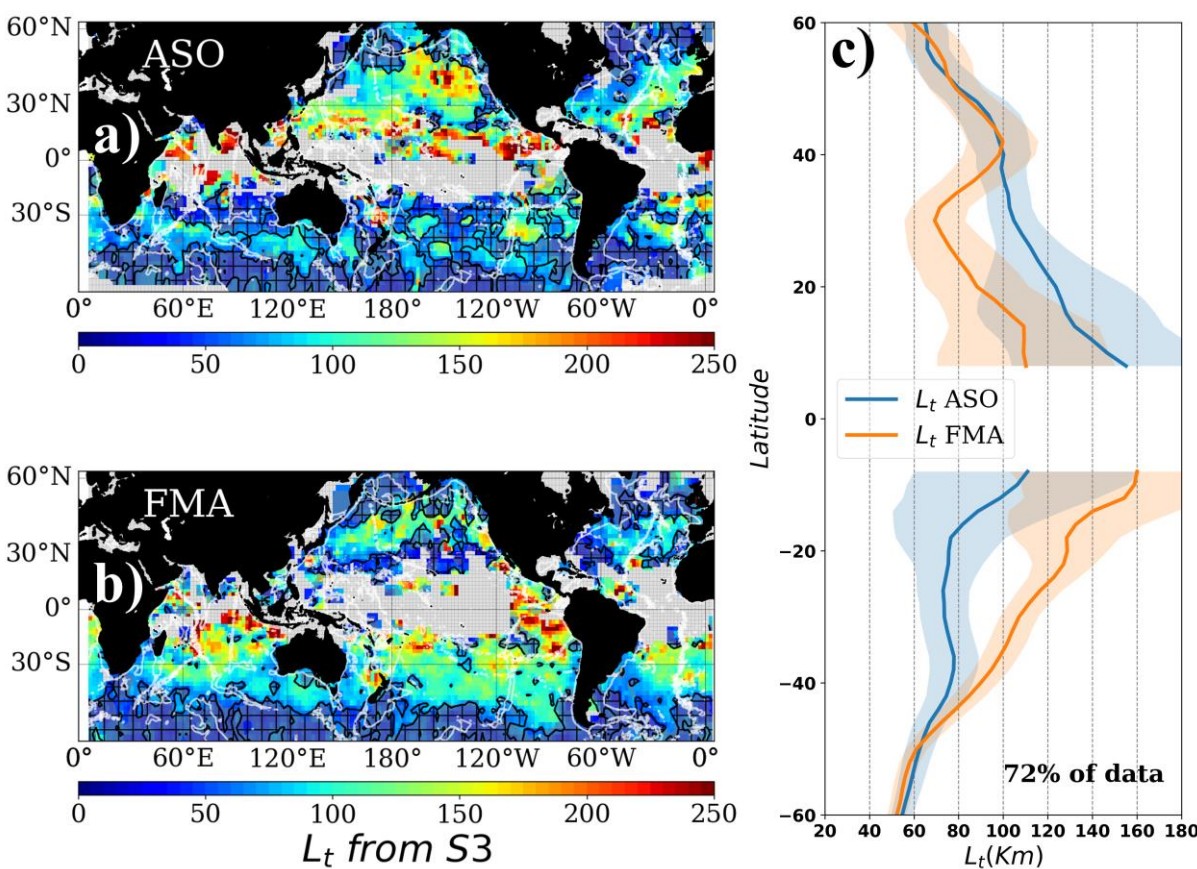

**Figure 11.** Same as Figure 10, but for Sentinel-3.

We note that our zonally-averaged results are only calculated in the non-masked areas and have partial coverage, but these first altimeter-based seasonal Lt modulation agrees with the modeling results reported by Qiu et al. (2018). The altimeter spectral-based seasonal Lt values are however longer than that reported in Qiu et al., (2018) from a modeled SSH, by 30 to 70 km (Bo Qiu, personal communication). Several key methodological differences could explain

the observed differences. While Qiu et al (2018) use either a spectral filter over SSH field, or a Helmholtz decomposition of the velocity field in order to determine Lt, we use the observed SSH and the change in spectral shape as a proxy for the boundary between large- and small-scale in the observed SSH spectrum. Our spectral approach, although straightforward, is coarser in comparison, given that the spectral shape analyzed (over the wavelength range of 605 interest) contains a mixture of large- and small-scale dynamics and residuals of imperfect instrumental corrections inherent to the satellite altimetry technique. The overall influence of these factors is accounted for by the uncertainty envelope that is generated from our statistical averaging, yielding an uncertainty of a few tens of km in some regions. Whereas Qiu et al. (2018) generates a precise separation between the high- and low-frequency parts of the 610 modelled SSH spectrum, by filtering the SSH signal using a thorough methodology based on the data-derived dispersion relation for higher dynamical modes and different tidal constituents up to $O_1$, which are not readily available for the altimetric observations.

Although our analyses show the possibility to partially diagnose the small-scale part of the SSH spectrum, a thorough diagnosis of the impact of the instrumental noise levels on the 615 methodology presented in this paper should be also carried out. This could be built around a series of Observing System Simulation Experiments that simulate the along-track observations, and also isolate the different contribution to the SSH energy spectrum. This is planned for future work.

**4.4 Implications for altimetric mapping and the SWOT mission**

There are two major implications for these spectral analyses results. The first is that the observable wavelength of all SSH signals above the instrument noise are limited to 60-70 km for Jason-3 and 50-70 km for S3-A (Figures 1 and 2). So, at present, any alongtrack altimetric studies addressing either balanced ocean dynamics or internal tides or internal gravity waves

will be limited to these spatial scales by this instrument noise level. Recent improvements in high-resolution 20/40 Hz processing techniques for the alongtrack altimetric signal aiming to improve the SNR of existing data show a reduction of the noise plateau in the order of 20% for 1Hz data (Tran et al., 2021, Quilfen and Chapron, 2021). This may improve the lower bound of our estimated OWL, as recent results using the latest SAR processing suggest (Moreau et al., 2021; Pujol et al., 2023).

The performance of the upcoming SWOT mission, embarking a new generation of altimeter technology, anticipates more than one order of magnitude of noise level reduction compared to current 1Hz Jason observations (Fu & Ubelmann, 2014). Refined estimates for the SWOT SNR including realistic wind-wave effects on the interferometric technology anticipate an observability ranging from around 15 km in wavelength at low latitudes to around 30-45 km in wavelength towards the poles, with an important longitudinal dependence (Wang et al. 2019). This will greatly extend our capacity to estimate the smaller-spectral slope, especially at mid to high latitudes.

The second implication is for the Lt intercept wavelength. This wavelength value is critical when calculating geostrophic currents from altimetric SSH slopes. If the smaller-slope at shorter wavelengths is indeed dominated by ageostrophic dynamics and internal tides or internal gravity waves, these contributions will induce large errors in the geostrophic current calculation. In our analysis, we have not included any recent corrections to remove the phase-locked internal tide from the alongtrack altimetry data. Separate tests (not shown) indicate that these coherent internal tide corrections have a minor impact on our Lt results, since the non-phase-locked internal tide remains, and the cascade of internal tide energy to smaller scales contributes to the small-scale slope over similar wavelengths. So this regionally varying Lt value needs to be taken into account when choosing the appropriate spatial scales to calculate geostrophic currents: either for the cross-track geostrophic currents from the

alongtrack SSH slopes, or when mapping the alongtrack data onto a regular grid. Although our alongtrack Lt statistical values are not available globally, due to the limitations with the current generation of altimetric noise, the limited values are consistent with the modeling results of Qiu et al. (2018). This suggests that 1) these changes of slope, predicted by the models, are observable in limited regions with today's altimetry missions, and 2) that the global modelled Lt values can be used as a good estimation of the appropriate spatial scales for separating balanced motions for geostrophic current calculations with altimeter data.

Our results indicate that at low latitudes, the intercept wavelength remains large (100-150 km) suggesting that the changes in the spectral slope will be well observed in two-dimensions by the future SWOT mission with its reduced noise level. The estimated observability at high latitudes, particularly in the ACC could still be a challenge for diagnosing the transition from geostrophic to non-geostrophic circulation regimes from SWOT observations alone, unless noise-reduction techniques are also applied to SWOT data. Wang et al. (2019) estimate the observed wavelengths for SWOT in the ACC to be between 30 and 45 km wavelength, with an important longitudinal dependency. Our very limited alongtrack estimates in this region indicate that the spectral slope break should occur in the 40-60 km wavelength range on average, as do the modeling estimates from Qiu et al (2018). *In situ* observations from the Drake Passage report that half of the near surface kinetic energy between 10 and 40 km wavelength is accounted for by ageostrophic motions (Rocha et al., 2016a), likely dominated by inertia-gravity waves. Our estimates also reveal an inherent geographical variability of the intercept wavelength, suggesting a localized dependence of the different dynamical regimes around the ACC that was also observed by Wang et al. (2019) for the region. This implies that the observability in the ACC will not be a constant threshold but rather a pattern dominated by localized and seasonal variability.

**Appendix A**

Several studies have reported the effect of sea and swell on SAR-mode acquisition satellites
such as Sentinel-3A (Moreau et al., 2018; Rieu et al. 2021; Moreau et al., 2021), highlighting
that SAR-specific processing methods result in a PSD signature at high wavenumbers that
deviates from the expected random thermal noise. The expected signature of a random signal
in the high wavenumber part of the spectrum is a characteristic flat plateau or "white noise"
(in the case of 1Hz SSH, this concerns wavelengths shorter than 30 km wavelength). In the
case of Sentinel-3A, SSH data in this part of the spectrum exhibits a slightly slanted shape or
"red noise" plateau (Figure A1a and b, dashed lines).

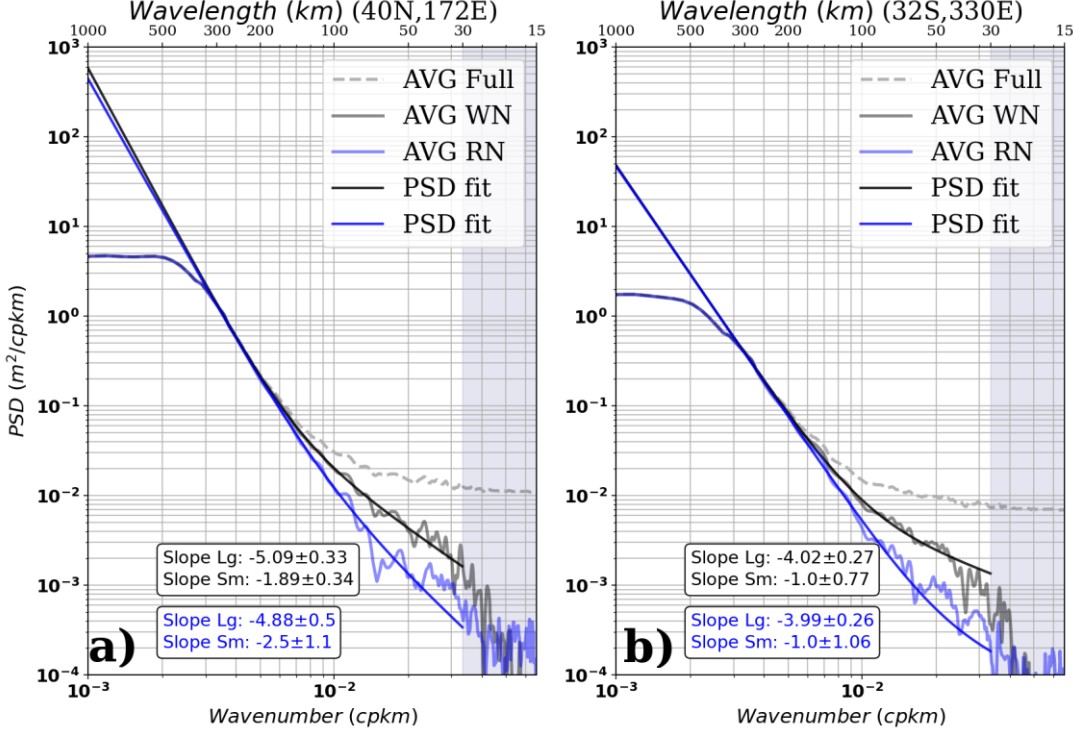

**Figure A1.** Result of the double fit algorithm over the denoised S3A data for the white
(black-WN) and red (blue-RN) noise model for a 12°x12° box over the North Pacific (a) and
the South Atlantic (b). Considering the error associated with the optimal fit analysis, the
spectral slopes obtained for either the black or blue curves are not significantly different.

Following Xu and Fu (2012), in the present paper we analyze the shape of the unbiased SSH
spectrum and therefore we assume that the thermal noise signature that dominates the SSH at

wavelengths shorter than 30 km is essentially a white-noise plateau. We perform a sensitivity test on the effect of using a red-type noise plateau rather than a white-type plateau for wavelengths between 15 and 30 Km wavelength to compute the unbiased S3A spectra. Figure A1 shows the result of using the two different functions as the approximation for the noise plateau over selected regions, Figure A2 shows the spatial distribution of the mesoscale and small-scale spectral slopes resulting from the observed PSD denoised using a red-type noise. Results show that the mesoscale spectral slope is not significantly different between both methods of denoising (Fig. A1, Figures A2a and A2c) and that the same geographical patterns can be observed in the two cases (Figures 5a and A2a). The red noise estimate reduces the PSD levels particularly at smaller wavelengths, with a small impact on the small-scale spectral slope. These small-scale spectral slope values vary between 1.5 and 2.5 for all latitudes equatorward of $40°$ (Fig. A2b), and the most important differences with the white-noise unbiased case are observed between $30°$ and $40°$, with the latter showing steeper slopes (eg Figure A1a), albeit falling inside the uncertainty envelope on average (Fig. A2c). More evident differences between the two methods arise at high-latitudes where the small-scale spectral slopes for the red-noise unbiased PSD are lower than for its white-noise counterpart (Figures 5b and A2b). The uncertainty associated with the small-scale spectral slope is also higher for the red-noise unbiased PSD, resulting from the additional energy that is subtracted by the denoising process in comparison with a white-plateau. This is illustrated by the different energy levels observed for the blue and black curves in Figure A1. Therefore, the uncertainty associated with the spectral slopes at small scales are higher for the red-noise unbiased spectra than for the white-noise unbiased spectra (note that the shaded areas in Fig A2b are more important than in Fig. 5b).

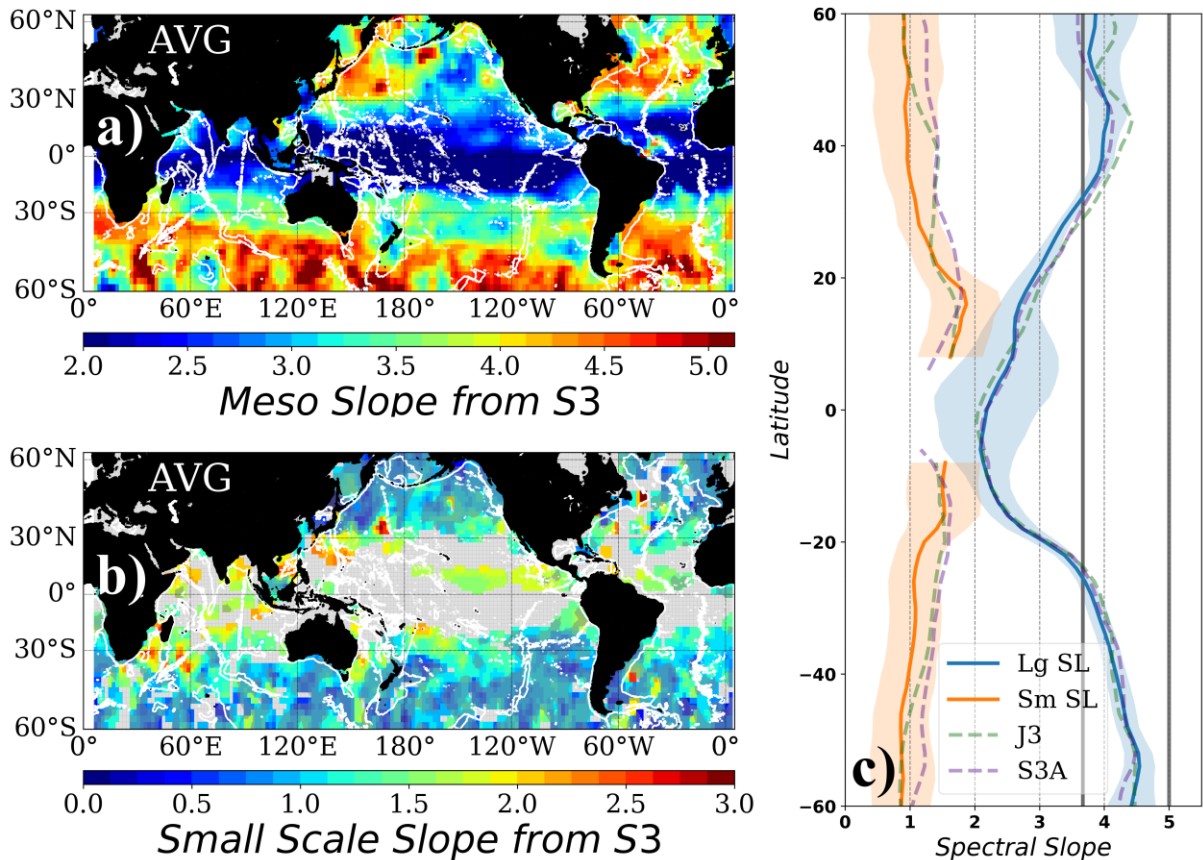

**Figure A2.** (a) Mesoscale and (b) small-scale spectral slopes fitted to the observed unbiased PSD from Sentinel-3A. The noise function used to unbiased the full PSD corresponds to a red-type noise function. (c) Zonal average of the mesoscale and small-scale spectral slopes in (a) and (b). Vertical gray lines denote the $k^{-11/3}$ and $k^{-5}$ spectral slope values. Gray shading in (b) corresponds to zones where the uncertainty associated with the slope estimate is higher than 40% of the slope value. Blank areas in (b) correspond to zones where the double slope model does not describe the observed shape of the spectrum. Dashed green and purple lines in (c) correspond to Jason-3 and Sentinel-3A zonal averages from Figures 4c and 5c respectively. White contours represent the topography at 3000 m depth.

**Appendix B**

In the following we present the seasonal results of the meso- and small-scale spectral slopes, and for the intercept wavelength at three regions. This analysis complements the discussion presented in Section 4.3.

The seasonality of the spectral slope observed by the two missions (Figure B1) shows a distinctive pattern around the average values (Figures 4 and 5), with maximum values in the vicinity of highly energetic regions (e.g. Kuroshio extension, Gulf Stream, ACC), and rather shallow slopes in the intertropical band. In agreement with recent studies (e.g. Dufau et al., 2016; Vergara et al., 2019), slope values during summer are higher than during winter, indicative of a contrasting dynamical regime during both seasons. These changes are essentially related to the seasonal changes in the vertical structure of the upper ocean, which would allow the development of vigorous small eddies during winter months (Rocha et al., 2016b) that tend to flatten and reduce the mesoscale slope of the spectrum (Figures B1c). This enhanced late-winter small-scale energy may be transferred up-scale to larger-scale eddies in summer at mid-latitudes (eg Sasaki et al., 2014), increasing the energy levels in the mesoscale wavelength range during summer months. There is nearly null seasonal modulation in the equatorial band (10°S-10°N), where the seasonal changes in stratification are rather limited.

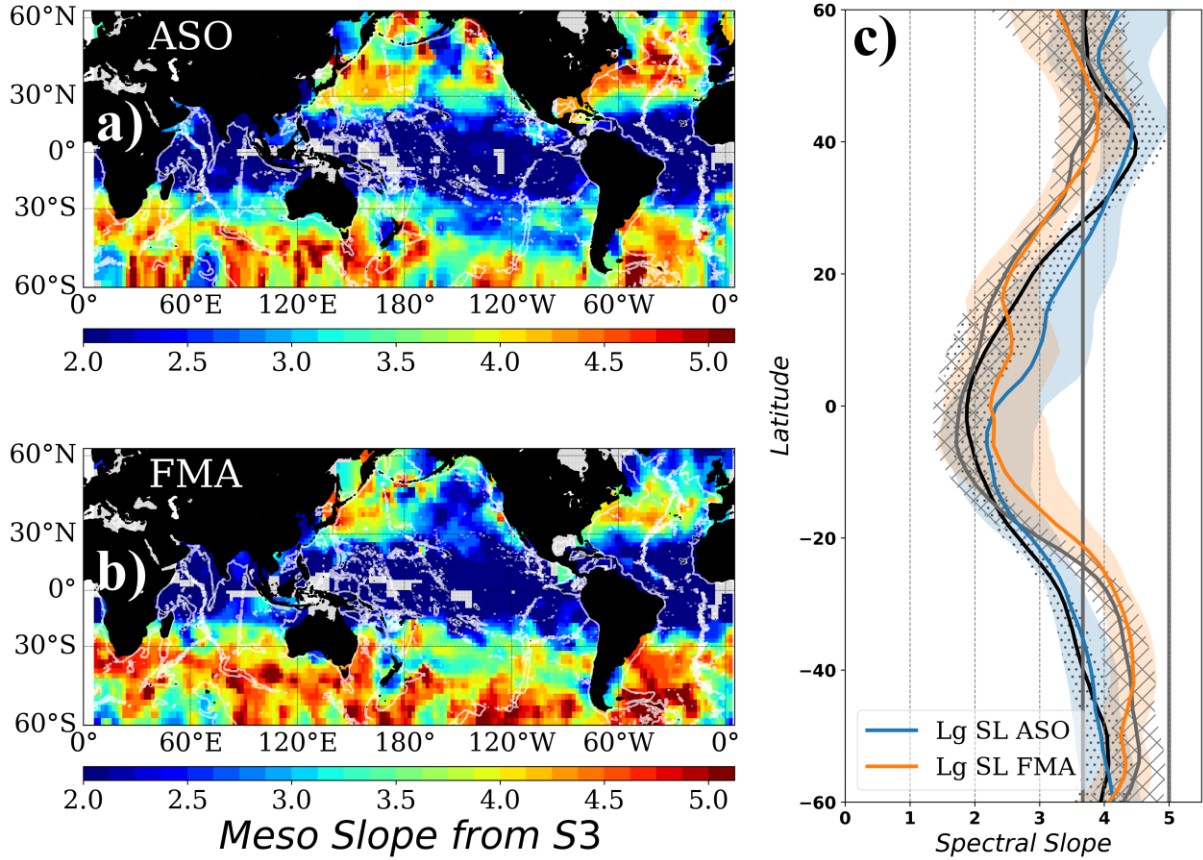

**Figure B1.** Mesoscale spectral slope averaged during (a) August-September-October and (b) February-March-April for Sentinel-3. (c) Zonal averages of (a) and (b). ASO (black line, dotted shading) and FMA (gray line, hatched shading) zonal averages for Jason-3 are also plotted in (c). Thick gray lines in (c) denote the spectral slope values of 11/3 and 5. White contours in (a) and (b) represent the topography at 3000 m depth.

Seasonal small-slope values follow a different pattern compared to the mesoscale spectral slopes, with zonally-averaged higher slopes during winter compared to summer (Figure B2c). These zonal statistics are dominated by the western-boundary regions of the Kuroshio, the East Australian Current, the Gulf Stream and the Brazil-Malvinas Confluence regions. There are smaller seasonal differences over low eddy energy regions such as the Southeastern Pacific. The winter-summer asymmetry suggests a change in the circulation variability, tied to the distinctive annual cycle observed in the 10-100 Km wavelength range (Callies et al., 2015;

Qiu et al., 2017) and may indicate that a wintertime energization of the small scales related to mixed layer instabilities (e.g. Lawrence and Callies, 2022), echoes on the small-scale spectral slopes observed here. As for the rest of the paper, we restricted our analyses and zonal averages to the zones where the spectral slope error is lower than 40%.

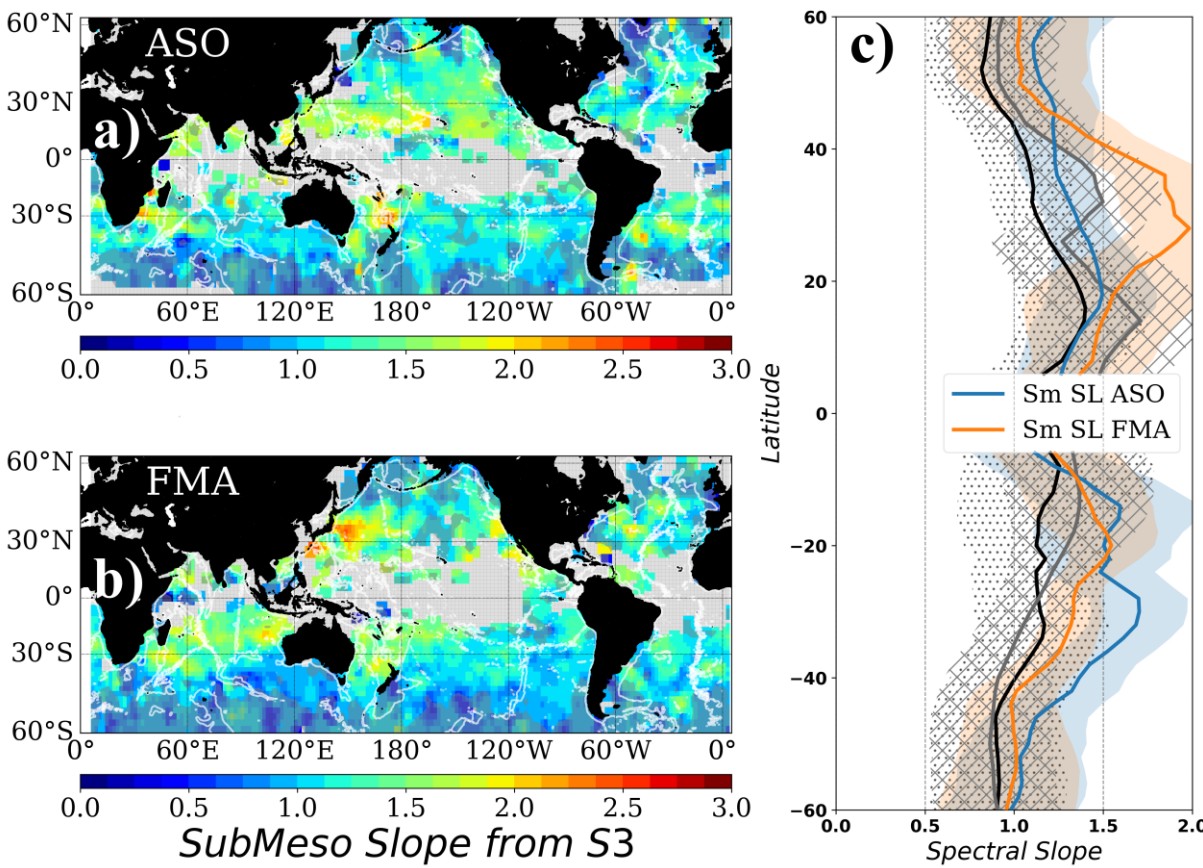

**Figure B2.** Small-scale spectral slope averaged during (a) August-September-October and (b) February-March-April for Sentinel-3. Gray shading in (a) and (b) corresponds to zones where the uncertainty associated with the slope estimate is higher than 40% of the slope value. Blank areas in (a) and (b) correspond to zones where the double slope model does not describe the observed shape of the spectrum. (c) Zonal averages of the valid pixels in (a) and (b). ASO (black line, dotted shading) and FMA (gray line, hatched shading) zonal averages for Jason-3 are also plotted in (c). White contours in (a) and (b) represent the topography at 3000 m depth.

In addition to the seasonal modulation of the intercept wavelength illustrated by Figures 10 and 11, we analyzed the statistical distribution of Lt over three western boundary regions (Fig. B3). Longer intercept wavelengths are observed during summer months compared to winter (20-35 km longer, significant at 95%), with differences between the two sets of observations of around 10 to 12 km.

A detailed analysis using the latest altimetric data available with lower noise levels (Moreau et al., 2021) over a longer time-series is planned for future work, including comparisons against *in situ* measurements and results from recent literature (e.g. Qiu et al., 2017).

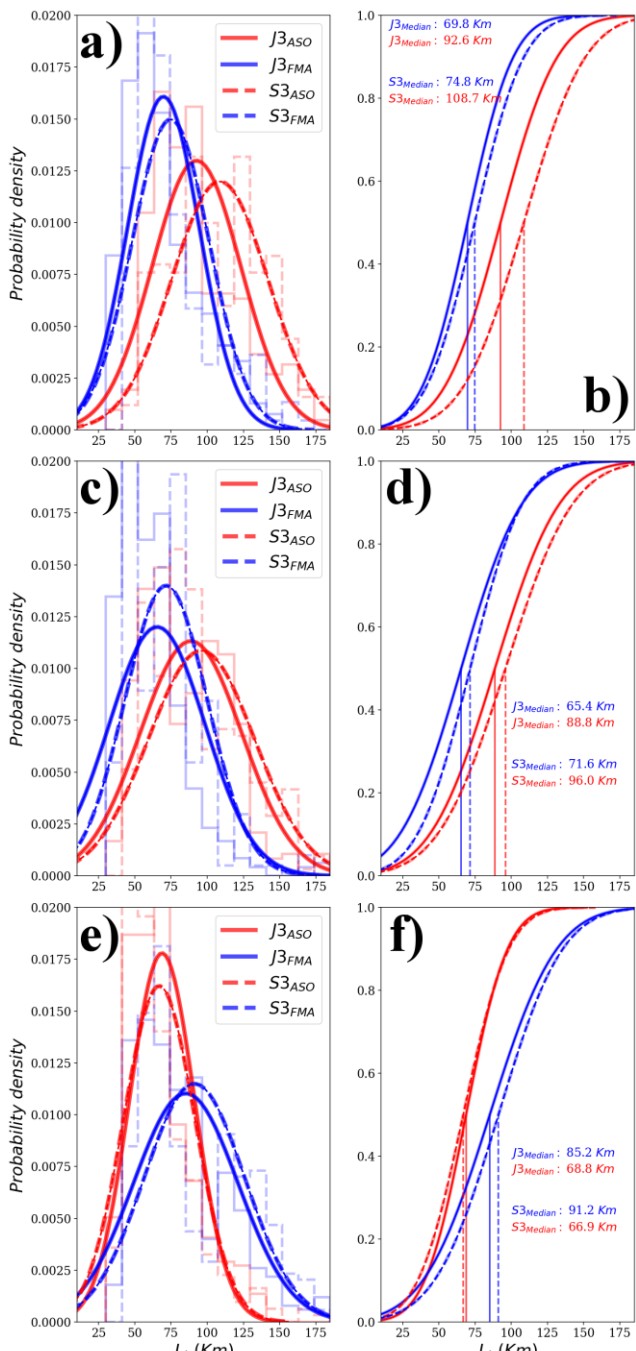

**Figure B3.** Histogram and cumulative distribution function for the values of the intercept wavelength observed during August-September-October (ASO) and February-March-April (FMA), for Jason-3 (full line) and Sentinel-3 (dashed line) over three regions: (a), (b) North Western Pacific (20-45°N; 135-170°E); (c), (d) North Western Atlantic (25-45°N; 280-320°E); South Western Atlantic (55-25°S; 290-320°E). Only the pixels with seasonal differences significant at 95% confidence were considered.

**Data availability**

The altimetry data used in the present paper is fully available at AVISO's website: https://www.aviso.altimetry.fr/en/home.html .

**Author contributions**

OV, RM and M-IP conceived the methodology used to analyze the sea surface height spectra and its shape changes. The analyses presented in this paper are the collective effort of all the authors. OV wrote the manuscript in close collaboration with RM. M-IP, GD and CU provided altimetry data expertise and critical advice on the methodology.

**Competing interests**

The authors declare that they have no conflict of interests.

**Acknowledgments**

The Jason-3 and Sentinel-3A data were obtained from the Geophysical Data Record (GDR) available on the AVISO website (https://www.aviso.altimetry.fr). The Mean Sea Surface Model CNES_CLS_2015 is also available on the same website. The authors would like to acknowledge the support from the French Space Agency (CNES) via the French TOSCA program to conduct the present study. Constructive comments made by two anonymous reviewers helped improve an early version of the manuscript.

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
