# Peer review of "Global submesoscale Transition Scale estimation using alongtrack satellite altimetry"

_EGUsphere, 2022_

## Author Comment (AC1)

Response to Reviewer #1

*Referee comment on "Global submesoscale diagnosis using alongtrack satellite altimetry" by Oscar Vergara et al., EGUsphere, https://doi.org/10.5194/egusphere-2022-1073-RC1, 2022*

**Minor comments**

1. The global pattern of Lt (Fig. 6) is similar to that in Fig4 of Qiu et al. (2018) using a high-resolution simulation. However, most values of Lt appear to be relatively larger than those in Qiu et al. (2018). In addition, the Lt in this study is also relatively larger than that of the ADCP observations in the Western Pacific in Qiu et al. (2017). Could you provide the reason or discussion about this issue?

Response: The reviewer is correct. Our results show larger Lt values than Qiu et al. (2018) results from an OGCM and that of Qiu et al. (2017) from 12 years of ADCP observations. In a similar manner as Qiu et al. in both their papers, our work aims to tease out a regime change on the surface ocean dynamics based on direct SSH observations. However, there are several key methodological differences between our work and that of Qiu et al.'s when analysing the energy content in SSH signature. While Qiu et al (2018) use either a spectral filter (over SSH field) or the Helmholtz decomposition (for u and v; analogous to Qiu et al. (2017)) to determine Lt, we use the observed SSH and the change in spectral shape as a proxy for the boundary between large- and small-scale in the observed SSH spectrum. Our approach is therefore more "crude", given that the spectral shape that we analyse (over the wavelength range of interest) contains a mix of large- and small-scale dynamics and residuals of imperfect instrumental corrections inherent to the satellite altimetry technique. The overall influence of these factors is accounted for by the uncertainty envelope that is generated from our statistical averaging, which yields an uncertainty around a few tenths of km in some regions. On the other hand, Qiu et al. (2018) generates a precise separation between the high- and low-frequency parts of the analysed SSH spectrum, by filtering the SSH signal using a thorough methodology based on the observed dispersion relation for higher dynamical modes and different tidal constituents up to $O_1$.

Qiu et al. (2018) also note that the Lt estimates differ between the SSH-based estimates and their Kinetic Energy-based estimates, with the SSH-bases estimates being larger by 20 to 100 km inside the 40° latitude band (Figure 12, Qiu et la., (2018)), attributing this effect to the low contribution of near-inertial motions to the SSH signal (as opposed to u and v), which could also have an impact on our estimates and how they compare to the very short Lt values reported in Qiu et al. (2017), using a historical ADCP time-series over the Northwest Pacific. In this matter, Qiu et al. (2018) note that their Lt estimates are larger than those using the *in situ* data in Qiu et al. (2017), and argue that the time-series lengths could also have an impact over the average Lt estimates (1 year versus 12 years in this case). This could also play a role on our Lt estimates.

All these considerations open several scientific questions that we intend to answer in future works. We plan to evaluate the impact of the instrumental noise and the different components of the SSH spectrum on our estimates by performing our analysis over simulated SSH fields (from mitGCM for example).

Overall, the Lt estimates presented in our work are larger than those of Qiu et al. (2018) by 30 to 50 km on average (Figure 7 of our submitted paper and Figure 12a of Qiu et al., (2018). Our

seasonal estimates (Figures 10 and 11) also differ from the SSH-based Lt estimates of Qiu et al (2018) by 30 to 50 km on average (Bo Qiu, personal communication).

For clarity, we expanded the discussion in the revised version of the manuscript to include some of the elements mentioned above (Section 4.3).

2. L66: Please add a reference (Lawrence and Callies 2022).

R.: The reference was added as suggested by the reviewer.

3. L120-121: Could you explain a little bit more about the distribution of noise level? Are the noise levels high in the regions around the Gulf Stream, Kuroshio Extension, and ACC? L123-124: Does the noise level increase simply increase from the equator to the poles?

R.: The noise levels observed for both satellites indeed show local maxima in the vicinity of the Gulf Stream, Kuroshio extension and the ACC, related essentially to local geophysical signals such as rain cells and more importantly the local wind wave field. Despite the relatively higher noise levels observed in these regions, the mesoscale signal is also strong and therefore the signal to noise ratio is good. This is reflected by the observability wavelength, showing local minima around these regions. The case of the ACC is more complex; although the SSH variability is very energetic here, it is the least favourable regions in terms of noise levels, as the "spotty" observability patterns suggest (Figures 1b and 2b).

We modified the text in the revised version of the manuscript for clarity.

4. L261: "Mesoscale spectral slope" should be "3.1.1 Mesoscale spectral slope".

R.: This was modified as requested by the reviewer.

5. L309: "Small-scale spectral slope" should be "3.1.1 Small-scale spectral slope".

R.: This was modified as requested by the reviewer.

6. Caption of Fig.4: Please clarify the shadings in Fig. 4c.

R.: The figure caption was modified for clarity.

7. Caption of Fig.5: Is the figure the same as Fig.4, but S3A? And dashed gray lines in (c) correspond to Jason-3 zonal averages from Figure 4c.

R.: The reviewer is correct, Figure 5 is the same as Figure 4 but for Sentinel-3 and the gray dashed lines correspond to the zonal averages from Figure 4c. The caption was modified for clarity.

8. L325 "as well as an increase in the uncertainty": Does it mean that the blank zone corresponding to not describing the small-scale slope increases toward the poles?

R.: This sentence refers to the increase in the uncertainty around the computed values for the small-scale spectral slopes as the latitude increases (Figure 4b and 5b). This does not necessarily mean that the zones not describing the small-scale slopes increase polewards, but rather that the dispersion around the average values increases (i.e. the uncertainty). The sentence was modified to clarify this point.

9. L344: The tropical instability waves distribute between 10S and 10N.

R.: The text was corrected.

---

## Author Comment (AC2)

Response to Reviewer #2

*Referee comment on "Global submesoscale diagnosis using alongtrack satellite altimetry" by Oscar Vergara et al., EGUsphere, https://doi.org/10.5194/egusphere-2022-1073-RC1, 2022*

**Major comments**

1. It has been shown that SSH and its spectrum have a clear seasonality. It is worth a separate section to look into the same set of diagnoses but considering their seasonality, i.e., of the submesoscale signals as well as the transition scales and spectral slopes.

Response: Following the suggestion by the reviewer we decided to add a section documenting the seasonal variability observed in the meso- and small-scale spectral slopes as well as Lt. However, the robustness of the seasonal results is heavily conditioned by the time series length, the observability and uncertainty criteria (same criteria used to report the average results presented in the previous version of the paper). With three years' worth of data, the seasonal averages are performed over a period equivalent to 9 months (3-month average for every season). As a result, we do not observe a global coverage of significant seasonal differences (at 95% confidence). We decided nevertheless to include the seasonal Lt results in the discussion section (Section 4.3) and the spectral slopes as supplementary material (Appendix B).

**Minor comments**

1. Line 217: "the local observability wavelength is low" can be reworded to "the local observable wavelength is short".

R: The sentence was rearranged.

2. Line 141: change "the regionally average" to "the regionally-averaged".

R: The misspell error was corrected.

3. Line 143: do you mean "spectral slope variation" by "spectral variance"?

R: The sentence was modified for clarity.

4. Line 145: provide an quantitative description of "changes abruptly"

R: The sentence was modified for clarity.

5. Line 153: define "x"

R: The sentence was modified.

6. Line 154: add "in the log-log coordinate" after "two straight lines"

R: The sentence was modified.

7. Line 160: by "coherent", did you mean "consistent"?

R: We meant "continuous", as in a continuous function. Applying a simultaneous fit assures a continuity of the model used to reproduce the observations, as opposed to successive individual fits where the sum of the models would not reproduce the observations as each individual model is minimized over a certain wavelength range. The sentence was modified to reflect this reasoning.

8. Line 161: replace "average regional" with "regionally-averaged"

R: The sentence was modified.

9. Line 170: "least-squares" should be "least-square"

R: The sentence was modified.

10. Line 217: replace "short" with "small" as it describes wavelength

R: The sentence was modified.

11. Line 201: delete "values"

R: The sentence was corrected.

12. Line 339: Did you mean "significant"?

R: The misspell error was corrected.

13. Line 435: replace "lowest" with "shortest"

R: The sentence was modified.

14. Line 543: change "of critical importance" to "critical"

R: The sentence was modified.